# Explainable depth-wise and channel-wise fusion models for multi-class skin lesion classification

Humam AbuAlkebash[1]*, Radhwan A. A. Saleh[2,3]*, H. Metin Ertunç[4]

**1** Department of Mechatronics Engineering, Palestine Polytechnic University, Hebron, Palestine, **2** Faculty of Engineering and Information Technology, Taiz University, Taiz, Yemen, **3** Department of Software Engineering, Kocaeli University, Kocaeli, Türkiye, **4** Department of Mechatronics Engineering, Kocaeli University, Kocaeli, Türkiye

* p6378@ppu.edu.ps (HA); radhwan.saleh@taiz.edu.ye (RAAS)

## Abstract

The clinical adoption of deep learning in dermatology requires models that are not only highly accurate but also transparent and trustworthy. To address this dual challenge, this study presents a systematic investigation into deep feature fusion, exploring how to effectively combine complementary representations from diverse neural network architectures. We design and rigorously evaluate six distinct fusion models, first investigating depth-wise and channel-wise strategies for integrating features from powerful Convolutional Neural Network (CNN) backbones, and subsequently incorporating the global contextual awareness of Vision Transformers (ViTs). Evaluated on the challenging 7-class HAM10000 dataset, our optimized architecture achieves a weighted average Precision, Recall, and F1 score of 90%, demonstrating superior diagnostic performance. Crucially, our comprehensive explainable AI (XAI) analysis using Grad-CAM and SHAP reveals that the fusion strategy directly dictates the model's clinical interpretability; our most effective models learn to base their predictions on salient dermatological features, such as border irregularity and color variegation, in a manner that aligns with expert reasoning. This work provides a robust framework and valuable architectural insights for developing the next generation of high-performing, clinically reliable, and transparent AI-powered diagnostic tools.

## Introduction

Skin cancer is one of the most common types of cancer in the current decade [1]. It takes place due to an uncontrolled increase of abnormal skin cells. This leads to the formation of malignant tumors that affect millions of people worldwide, regardless of their age or skin type. The principal cause is unprotected skin exposure to UV rays [2–4]. Skin cancer is primarily categorized into two main types: non-melanoma and melanoma [5]. Melanoma is a hazardous, uncommon, and deadly form of malignant skin cancer. While it represents 5% of skin cancers, it exhibits a higher mortality rate.

**Data availability statement:** All data records of the HAM10000 dataset are deposited at the Harvard Dataverse.

**Funding:** The author(s) received no specific funding for this work.

**Competing interests:** The authors have declared that no competing interests exist.

Although it can appear anywhere on sun-exposed parts of the body, it commonly affects regions such as the hands, cheeks, neck, and lips [6]. The remaining 95% are nonmelanoma skin cancers, like basal cell carcinoma (BCC) and squamous cell carcinoma (SCC), which are generally less aggressive and have a lower likelihood of spreading to other parts of the body [7]. Generally, skin cancer is a serious health danger and is very common worldwide. The number of skin cancer cases annually increased by 55% between 2009 and 2019 [8]. Luckily, early diagnosis increases the likelihood of recovery for all types of skin cancer [9]. Furthermore, the survival rate can be increased to almost 90 percent with early detection [10,11].

Melanocyte mutations from long-term UV exposure can cause cells to create too much melanin, causing dark moles. Moles and blemishes can become malignant tumors, such as melanoma, that spread and damage other bodily regions. If left unchecked, they can grow unmanageable and endanger health. Fortunately, early detection and treatment of melanoma can greatly minimize morbidity and mortality. Numerous studies have shown that early melanoma detection improves prognosis and cure rates. Therefore, early identification and precise diagnosis are crucial to improve melanoma survival. Early detection and diagnosis can help patients and their families have a better future by minimizing disease incidence and severity [12].

Recently, many efforts have been undertaken to create computer-aided skin lesion analysis methods. ABCD (Asymmetry Borders-Colors-Dermatoscopic Structures) is a popular method for assessing asymmetry, border irregularity, non-uniform color, and dermatoscopic structures. Dermatologists utilize this rule to diagnose skin cancer. Manual skin diagnosis is difficult and burdensome for patients [13]. In actuality, melanoma images might present multiple challenges, such as low contrast, noise, and inconsistent boundaries. These challenges can prevent medical professionals from precisely diagnosing the disease in every case. The Melanoma Research Foundation asserts that the correct diagnosis of melanoma using the ABCD standards is not consistently feasible across all images [14]. Furthermore, this method necessitates the involvement of highly skilled dermatologists to minimize the risk of incorrect diagnoses, which may lead to a significant incidence of false positives and false negatives. Moreover, the diagnosis of skin cancer growth has traditionally relied on histological analysis of a sample substance, a cycle that can be labor-intensive and time-consuming. Besides, changes in interpretation among specialists may lead to inequality in diagnosis [12].

Given these limitations in the melanoma diagnosis process, there is an increasing demand for intelligent and automated tools to help dermatologists in making the diagnosis. This has led to the development of algorithms for the automatic identification of melanoma, specifically emphasizing those capable of efficiently extracting information from dermoscopic images. These promising algorithms have attracted considerable attention in recent years because they can help dermatologists screen for melanoma. This improves the overall accuracy and efficiency of melanoma detection. Progress in computer vision and Deep Learning (DL) have opened up new prospects for automated skin cancer diagnosis.

In the literature, skin cancer detection has been achieved with competitive results utilizing Convolutional Neural Networks (CNNs), Vision Transformers (ViTs), and

attention mechanisms [15,16]. Beyond dermatology, recent architectural advances in broader medical imaging domains have demonstrated the superior efficacy of integrating convolutional backbones with hierarchical attention mechanisms [17–19]. For instance, novel frameworks utilizing pyramidal attention and feature-summarized networks have achieved state-of-the-art results in neuro-oncology [17,19] and pulmonary pathology [18] by effectively capturing multi-scale dependencies. These studies collectively highlight a paradigm shift towards hybrid architectures that balance local feature extraction with global contextual awareness, a principle that drives the fusion strategy proposed in this study. Nevertheless, it has been observed that the majority of studies failed to incorporate Explainable Artificial Intelligence (XAI) techniques to provide transparency and reliability to AI solutions. XAI methodologies are important to know how AI models make decisions, especially in healthcare. XAI is a big help with this, and it is crucial for dermatologists to feel confident using AI to help treat patients [20]. This gap in the literature is covered in this study by integrating XAI into deep learning models.

This research presents an Explainable AI fusion model that integrates Vision Transformers with pre-trained CNNs. The aim is to utilize the two different model architectures' advantages in order to enhance the efficacy of medical diagnosis. Multiple fusion procedures are examined to identify the most effective approach for integrating feature representations. The study incorporates two XAI approaches -Grad-CAM and SHAP values- that make the model's decision-making process more transparent. This transparency allows medical experts to understand which regions of the image influenced the model's diagnosis. Fig 1 illustrates the several stages that comprise the proposed framework. This methodology attempts to develop more accurate, transparent, and reliable AI diagnostic software for use in clinical settings through the following key contributions:

1. An XAI depth-wise and channel-wise fusion model is proposed for the purpose of skin cancer detection.
2. An investigation into various fusion procedures is conducted to find the most effective fusion architecture.
3. Two explainable AI techniques -Grad-CAM and SHAP values- are integrated to provide robust insights into the model's decision-making process. Fusing the suggested fusion deep learning models with other explainable AI methods, like Grad-CAM and SHAP values, provides a comprehensive understanding of the model's decision-making process.

## Related works

This section discusses related studies employing different deep-learning models to diagnose skin cancer. These studies can be categorized into CNN-based, Vision Transformer-based, and hybrid models. Each category is assessed on the strengths and weaknesses of the methodologies.

### CNN-based models

CNNs are widely used in various medical imaging applications, including skin cancer diagnosis, due to their ability to automatically extract relevant features from images [21,22]. Aldwgeri et al. [23] introduced an average voting-based ensemble of Deep Convolutional Neural Networks (DCNNs) to enhance dermoscopy image classification. They employed multiple pre-trained models, including DenseNet121, ResNet50, VGG-Net, Xception, and InceptionV3. In the three-class classification problem, the model achieved an area under the receiver operating characteristic curve (AUC) of 0.891 which significantly affects its practical use since the algorithm was trained to diagnose only three out of nine skin lesions. Goceri [24] presented a capsule-based network that has achieved an accuracy of 95.24%, outperforming other Capsule Networks and CNN-based models. However, it demands higher computational time than CNN-based models due to the complexity of the proposed routing mechanism. Furthermore, the model's efficacy has not been evaluated on images of lesions on darker skin tones, constraining its applicability across various races and skin types. Mridha et al. [25], has utilized a CNN-based deep learning framework to diagnose skin cancer. The trial and error method was employed to identify the

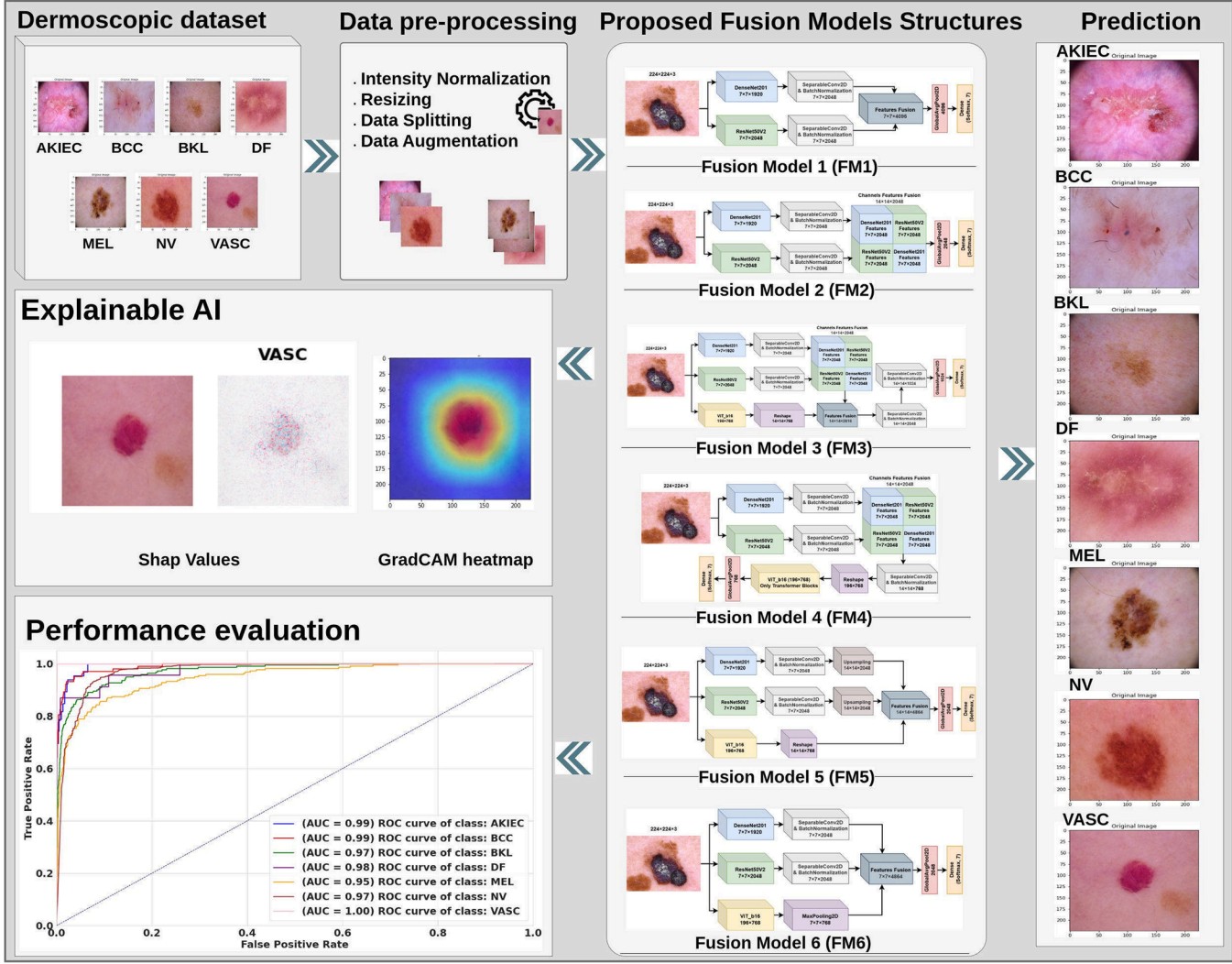

**Fig 1. Methodology of the proposed skin cancer detection framework.**

optimal parameters for the architecture. Three activation functions -ReLU, Tanh, and Swish- and two optimization algorithms -RMSprop and Adam- were utilized in the experiments to train the model. Their model achieved 82% classification accuracy in classifying skin lesions into seven classes using the HAM10000 dataset. Furthermore, they employed XAI methodologies (Grad-CAM and Grad-CAM++) to provide human-readable explanations of the model's decisions.

The authors in [26] have introduced what is called the Assist-Dermo methodology. They have used a depthwise separable CNN model derived from SqueezeNet as the basis for the Assist-Dermo classification framework. Compared with previously proposed DL models, SqueezeNet-Light shows substantial enhancements in performance and computational efficiency. This model was created for deployment on resource-limited devices including mobile and IoT platforms. It performs effectively in three-class scenarios, achieving an overall classification accuracy (ACC) of 94.5% and a 0.95 AUC. However, it has not been examined in seven-class scenarios, which could affect its robustness in real-world clinical settings. Furthermore, lesion segmentation in HSV color space and contrast enhancement are crucial to the system's performance thus preprocessing shortcomings can lower classification accuracy. Alhudhaif et al. [27] uses the

HAM10000 dataset across seven lesion classes to present a DL model that merges a CNN with a Soft Attention Module (SAM) for multiclass skin lesion classification to aid early skin cancer diagnosis. By inserting the SAM in the heart of the network, the features extracted are improved by focusing on the most important regions. SMOTE, ADASYN, RandomOverSampler, and Data Augmentation are used to solve the imbalance problem in the dataset, with SMOTE performing best. The balanced dataset employing SMOTE has achieved better results compared to the imbalanced dataset. This method improves classification accuracy and feature extraction, but it may overfit and require additional dataset validation. The paper solves the imbalance problem and shows that attention mechanisms and data balancing improve deep neural networks' multiclass skin lesion classification performance. However, the model exhibits an insufficient application of XAI and UQ methodologies, which raises questions regarding the interpretability and dependability of its predictions.

Lai et al. [28] combine optimization algorithms, CNN, and artificial neural networks (ANNs) to improve skin cancer detection. Kohonen neural network is improved by the Greedy Search Algorithm (GSA) to segment lesion regions accurately. CNN is used for feature extraction while ANN optimized by an Improved Gray Wolf Optimization (IGWO) is used for classification. On the ISIC-2016 and ISIC-2017 datasets, which address binary and multiclass classification tasks, the suggested framework has outperformed earlier methods by at least 0.5% with accuracy rates of 97.09% and 95.17%. Due to the larger population size required for the IGWO algorithm, this strategy requires more memory and computation but improves diagnostic accuracy and convergence time. Class imbalance and XAI are not addressed in this work.

Irfan Ali Kandhro et al. [29] introduces an enhanced VGG19 (E-VGG19) pre-trained deep learning model with max pooling and dense layers for binary classification of malignant and benign lesions. The approach uses the E-VGG19 model and other pre-trained models including ResNet152v2, InceptionResNetV2, DenseNet201, ResNet50, and InceptionV3 for feature extraction. At the same time, classic machine learning classifiers like SVM, KNN, DT, LR, and SVM are used for classification. Combining the E-VGG19 model with these classifiers improves classification accuracy. However, class imbalance issue, XAI, and UQ are not addressed in the paper. G. Akilandasowmya et al. [30] have involved the integration of deeply hidden features extracted using the Sand Cat Swarm Optimization with ResNet50 (SCSO-ResNet50) and ensemble classifiers. Feature selection is addressed using Enhanced Harmony Search (EHS) to solve high-dimensionality problems. ensemble classifiers that combine Naive Bayes, Random Forest, k-NN, SVM, and Linear Regression are used for classification. The proposed method surpassed state-of-the-art classifiers on benchmark datasets Kaggle skin cancer and ISIC 2019, achieving maximum accuracies of 93.568% and 94.238%, respectively. This approach handles real-time streaming and high-dimensional data well and improves prediction accuracy. However, optimization strategies increase computing complexity and resource consumption. Neither XAI nor class imbalance are addressed in the study.

Bong Kyung Jang and Yu Rang Park have proposed the Dynamically Expandable Representation (DER) incremental learning method [31]. The DER technique allows the model to adapt to new data and improve classification without retraining, improving medical diagnostics scalability and efficiency. Based on the HAM10000 dataset, the model has outperformed Xception, DenseNet-201, InceptionResNet-V2, GoogLeNet, and AlexNet with a classification accuracy of 80.88%. Additional validation using the ISIC 2019 dataset has verified the model's adaptability with an AUC of 0.911. The incremental learning method has helped the model to classify a variety of skin lesions by retaining and using earlier samples. The grad-CAM method is used to explain the model's reasoning process to clinicians. Limitations include unbalanced datasets influencing skin cancer performance and the need for ongoing updating as new diseases develop.

Transfer learning using EfficientNet models (B0 to B7) is used to classify multiclass skin cancer on the HAM10000 dataset [32]. A complete preprocessing workflow has removed hair from dermatoscopic images, augmented the dataset, and resized images to satisfy each model's input requirements. Intermediate complexity models (B4 and B5) outperformed simpler and more complicated variations with an F1 Score of 87% and a Top-1 Accuracy of 87.91%. This implies that model complexity does not always improve classification. It also has indicated that some skin cancer

groups generalized better than others. While the research acknowledges the high-class imbalance and evaluates models accordingly, it does not offer training methods to overcome it. XAI techniques are not employed to interpret model decisions.

Weicheng Yuan et al. present a semi-supervised skin cancer detection model using Self-feedback Threshold Focal Learning (STFL) to overcome the limitations of supervised learning approaches that require vast volumes of labeled data [33]. The STFL model uses a small number of labeled images and an enormous number of unlabeled medical images from the HAM10000 dataset to classify skin lesions into multiple classes. By dynamically modifying the selection threshold for unlabeled samples during training, the model filters out inaccurate data and uses focal learning to reduce class imbalance. In experiments with 500 annotated samples, the STFL model obtains 77% accuracy, 77% recall, 74.26% precision, and 74.62% F1 score. The model's performance improves with more labeled examples, demonstrating its scalability and efficiency. Dependence on unlabeled data quality and quantity and the requirement for validation across skin cancer classes and ethnicities to improve universal adaptability are limitations. Furthermore, XAI and UQ are not addressed in the paper.

More recently, attention mechanisms have become a primary focus for architectural innovation. Ghazouani [34] proposed the MuRANet, a hierarchical CNN that integrates multiple types of attention (e.g., multi-scale dilated, spatial, and channel attention) as residual connections. This approach aims to improve feature extraction in deep layers without information loss. On the HAM10000 dataset, the MuRANet achieved a highly competitive F1-score of 90% and an accuracy of 91%, underscoring the power of sophisticated, intrinsic attention architectures. Other works have focused on complex, multi-stage pipelines. Khan et al. [35], for instance, developed a framework that first fuses two CNNs for lesion segmentation, then uses a separate 30-layer CNN for feature extraction, and finally fuses and selects features before classification with an Extreme Learning Machine. This intricate approach yielded an accuracy of 87.02% on HAM10000. These studies highlight a trend towards either deeply integrated attention mechanisms or complex, multi-step processing pipelines to enhance the capabilities of CNN-based systems. On the other hand, Setiawan and Soewito [36] addressed the challenge of data imbalance not through architecture, but through a novel training methodology called Categorical Relation-Preserving Contrastive Decoupled Knowledge Distillation (CRCDKD). Applied to a DenseNet backbone, their method achieved an accuracy of 89.41% and a balanced F1-score of 83.39% on HAM10000, demonstrating that sophisticated training techniques can significantly boost performance, especially for minority classes.

Most recently, Banerjee [37] introduced the BCB-CSPA network, a sophisticated framework combining Bipartite Convoluted Blocks (BCB) with Condensed Semantic Perceptual Attention (CSPA). Evaluated on the HAM10000 dataset, this architecture demonstrated that refining features through pyramid multi-scale convolution significantly enhances diagnostic precision for challenging classes like AKIEC. Furthermore, their integration of Grad-CAM for interpretability aligns with the growing emphasis on Explainable AI in dermatoscopy. While BCB-CSPA illustrates the power of attention-driven CNNs, our work extends this direction by explicitly fusing convolutional features with Vision Transformers to capture long-range global dependencies that pure CNN architectures may miss.

Ebraheem Farea et al. [15] propose a novel hybrid AI framework that is specifically designed to address limitations such as the scarcity of medical datasets and the optimization of deep learning parameters in skin cancer detection. The framework consists of two critical steps: Initially, it compiles a comprehensive skin cancer dataset by integrating a variety of public datasets that cover a wide range of disease classes. Secondly, it meticulously optimizes the deep learning models using the Artificial Bee Colony (ABC) strategy, thereby effectively mitigating the adverse impacts of initial parameter randomness on AI performance. The hybrid framework's superior predictive capabilities are demonstrated by rigorous evaluations against eight prominent CNN models, including DenseNet121 and InceptionResNetV2. This hybrid framework makes a substantial contribution to the classification of multi-disease skin cancer and surpasses the capabilities of current AI models.

Table 1 summarizes these studies showing the key advantages and limitations in each study. Even though the CNN models proposed in the literature achieved high levels of accuracy that are comparable to dermatologists, their

**Table 1. Summary of recent studies in skin cancer diagnostics using deep learning techniques.**

| Reference | Year | Dataset | Model Proposed | Localization | Classes | Class Imbalance | Ensemble | XAI | Results |
|---|---|---|---|---|---|---|---|---|---|
| Aldwgeri et al. [23] | 2019 | ISIC 2018 | DenseNet121, ResNet50, VGG-Net, Xception, InceptionV3 | No | 7-class | No | Average Voting | No | AUC of 0.891, Accuracy of 80.1% |
| Ali et al. [32] | 2022 | HAM10000 | EfficientNet (B0-B7) | No | 7-class | No | No | No | F1 Score 87%, Accuracy 87.91% |
| Goceri [24] | 2023 | Only 805 images from HAM10000 | Capsule-based Network | No | Not mentioned | No | No | No | Accuracy 95.24% |
| Mridha et al. [25] | 2023 | HAM10000 | CNN-based Framework | No | 7-class | No | No | Grad-CAM, Grad-CAM++ | Accuracy 82% |
| Abbas et al. [26] | 2023 | Hybrid Dataset | Assist-Dermo (SqueezeNet-Light) | HSV Color Segmentation | 9-class | No | No | No | Accuracy 95.6%, AUC 0.95 |
| Alhudhaif et al. [27] | 2023 | HAM10000 | CNN with Soft Attention Module (SAM) | No | 7-class | SMOTE, ADASYN | No | No | Accuracy 75% |
| Lai et al. [28] | 2023 | ISIC-2016, ISIC-2017 | CNN with ANN, IGWO | Kohonen Network | Binary | No | No | No | Accuracy 97.09% (binary), 95.17% (multi) |
| Cirrincione et al. [38] | 2023 | ISIC-2017 | Vision Transformer (ViT) | No | 3-class | No | No | No | Accuracy 94.8% |
| Irfan Ali Kandhro et al. [29] | 2024 | - | E-VGG19 with various ML classifiers | No | Binary | No | No | No | Accuracy 88% |
| Akilandasowmya et al. [30] | 2024 | Kaggle Skin Cancer, ISIC 2019 | SCSO-ResNet50, Ensemble Classifiers | No | Binary | No | Voting | No | Accuracy 92.035% on Kaggle dataset, and 94.238% on ISIC 2019 |
| Bong Kyung Jang & Yu Rang Park [31] | 2024 | HAM10000, ISIC 2019 | Dynamically Expandable Representation (DER) | No | 6-class, 8-class | No | No | Grad-CAM | Accuracy 80.88%, AUC 0.911 |
| Weicheng Yuan et al. [33] | 2024 | HAM10000 | Self-feedback Threshold Focal Learning (STFL) | No | 7-class | Focal Learning | No | Grad-CAM | Accuracy 77%, Recall 77%, Precision 74.26% |
| Farea et al. [15] | 2024 | Hybrid Dataset | Hybrid AI Framework with ABC Optimization | Otsu's thresholding | 9-class | Augmentation, SMOTE | No | No | Accuracy 93.04% |
| Hatice Catal Reis [39] | 2024 | ISIC 2020 | MABSCNET with Ensemble | No | Binary | No | Fusion | Grad-CAM | Accuracy 92.74% |
| Desale & Patil [40] | 2024 | ISIC 2019 | Enhanced ViT with Preprocessing | Self-sparse Watershed Segmentation | 8-class | No | No | No | Accuracy 99.81% |
| Galib Muhammad et al. [41] | 2024 | HAM10000 | ViT with SAM | Segment Anything Model (SAM) | Binary | No | No | No | IOU 96.01%, Dice 98.14%, Accuracy 96.15% |
| Khosro Rezaee et al. [42] | 2024 | ISIC-2019, PH2 | Bi-branch CNN + Transformer with SAU | No | 8-class, 3-class | No | No | No | Accuracy 97.48% on ISIC-2019, 96.87% on PH2 |

decision-making process is difficult to comprehend, which restricts their explainability. Additionally, the computational cost of certain models, including the optimized SCD approach and the capsule-based network, restricts their applicability in environments with inadequate computational resources.

## Vision transformer-based models

Vision Transformers (ViTs) have recently gained popularity in image classification applications, such as skin cancer diagnosis, as a result of their ability to identify long-range features in images. For instance, Cirrincione et al. [38] have implemented a ViT-based framework to identify skin cancer. The images are appropriately processed, then the model is trained and tested using ViT. The model is assessed using the ISIC-2017 dataset and the results indicate an accuracy value of 94.8%. Hatice Catal Reis [39] introduces the Multi-Head Attention Block Depthwise Separable Convolution Network (MABSCNET), which integrates transformers with convolution layers. A hybrid architecture that integrates the MABSC-NET model with Ensemble Learning (EL) models that have been pre-trained on the ImageNet dataset is further described in the article. Its efficacy is evaluated by employing the ISIC 2020 dataset. The MABSCNET model has achieved an accuracy of 78.63%, while the ViT model has achieved 76.50% and the hybrid framework has achieved 92.74%. An optimal vision transformer strategy is presented by R. P. Desale and P. S. Patil in [40]. Utilizing techniques such as adaptive median filtering, Gaussian filtering, piecewise linear bottom hat filtering, and an enhanced gradient intensity method, the methodology entails pre-processing to ensure color consistency, remove hair artifacts, and reduce noise. The data is segmented using a self-sparse watershed approach, and features are extracted using the hybrid Walsh-Hadamard Karhunen-Loeve expansion technique. Finally, an enhanced vision transformer is implemented to classify skin cancer. In [41], Galib Muhammad et al. introduce a vision transformer-based approach that learns features by recording complex spatial and long-range dependencies through self-attention. The IOU and Dice values are 96.01% and 98.14%, respectively, as a consequence of the Segment Anything Model (SAM) being employed to segment malignant areas. Several pre-trained models are evaluated, and Google's ViT patch-32 model has achieved a low false negative ratio and 96.15% accuracy. Khosro Rezaee et al. [42] stand for a bi-branch parallel model for skin cancer classification that includes a Transformer module (TM), a self-attention unit (SAU), and a CNN. This model generates fine-grained features by cross-fusing local and global features. They introduce an optimized lightweight CNN (optResNet-18) that surpasses traditional CNNs such as ResNet-50 and ResNet-101, achieving 97.48% accuracy on the ISIC-2019 dataset and 96.87% on the PH2 dataset. The synergy between localization and classification has become a dominant theme in the latest hybrid model research. AbuAlkebash et al. [43] proposed a powerful two-stage framework where the YOLOv8 object detector is first used to precisely localize the lesion within the full dermoscopic image. The resulting cropped lesion is then passed to a Vision Transformer (ViT) for classification. This "detect-then-classify" strategy proved highly effective, achieving F1-score of 93% on the HAM10000 dataset. This work also underscored the importance of explainability by employing both Grad-CAM and SHAP to validate the model's decisions, a practice central to our own study. This recent studies illustrate that the frontier of skin cancer classification is being pushed by both architectural innovations, such as explicit localization, and advanced, data-centric training strategies.

Transformer-based models can accurately capture long-range dependencies in images, thereby improving the classification results of skin cancer issues. Nevertheless, the literature-proposed models need an extensive amount of training data to accomplish their optimal performance. Additionally, the transformer architecture requires extensive computational resources. Hybrid models offer superior performance by integrating the advantages of various methodologies. The model's ability to accurately classify skin lesions is improved by the integration of local and global variables. However, hybrid models are fundamentally more intricate and require a greater amount of computational resources than standalone models.

## Methodology

### Dataset and preprocessing

HAM10000 (Human Against Machine with 10000 training images) [44] is an open-source dataset essential for developing deep learning models in the classification of dermatological skin cancer. This dataset has a comprehensive set of dermoscopic images sourced from diverse patients and imaging techniques. Therefore, it reflects real-world clinical complexities. Our study comprised preprocessing steps such as data splitting and augmentation before model training. The publicly available HAM10000 dataset provides an important resource for skin cancer research, consisting of 10,015 high-resolution (600×450 pixels) dermoscopic images. It has seven distinct diagnostic classes, ranging from malignant lesions like melanoma (MEL) and basal cell carcinoma (BCC) to benign lesions such as melanocytic nevi (NV) and vascular lesions (VASC). HAM10000 offers a broad spectrum to investigate various types of skin cancer (MEL, VASC, benign keratosis-like lesions (BKL), actinic keratoses and intraepithelial carcinoma/Bowen's disease (AKIEC), BCC, dermatofibroma (DF), and NV). This diverse dataset is particularly valuable for developing and rigorously evaluating skin cancer detection models. The HAM10000 dataset combines dermatoscopic images from two distinct clinical sources to ensure diversity. These sources are the Department of Dermatology at the Medical University of Vienna, Austria, and a specialized skin cancer practice in Queensland, Australia. This multi-center collection addresses device heterogeneity by incorporating multiple imaging modalities, including the MoleMax HD digital dermatoscopy system, handheld DermLite dermoscopes (Fluid/DL3) coupled with digital cameras, and the high-resolution digitization of historical diapositives. Ground truth annotations were established through a rigorous hierarchy. Over 50% of lesions were confirmed via histopathology, while the remainder were validated through expert consensus, in-vivo confocal microscopy, or long-term clinical follow-up (1.5 years). This diverse acquisition protocol ensures the model is trained on data reflecting real-world clinical variability. The distribution of the HAM10000 dataset is illustrated in Table 2. As a result, the HAM10000 dataset is well-suited for research focusing on multiclass skin lesion classification and early skin cancer detection. Prior to model training, all images in the dataset underwent a standardized preprocessing pipeline to prepare them for deep learning models. The original high-resolution images were first resized to a uniform dimension of 224×224 pixels. This step is essential to match the fixed-size input requirements of the selected deep learning backbones. Following resizing, we performed intensity normalization. Specifically, the pixel values for each color channel, originally in the integer range of, were scaled to a floating point range of [0.0,1.0] by dividing by 255. This normalization is a standard and critical step that ensures numerical stability during training and facilitates efficient convergence of the model. To facilitate effective model training and unbiased evaluation, we randomly split the HAM10000 dataset by class into training (80%) and validation (20%) sets [45]. This stratified splitting strategy ensures a consistent class distribution across both sets, as detailed in Table 2.

### Data augmentation

The HAM10000 dataset, typical of real-world medical data, presents a significant class imbalance challenge. As shown in Table 2, the distribution of skin lesion types is skewed, with "Melanocytic nevi" (NV) representing a substantial 66.95%

**Table 2.** Class distribution in HAM10000 dataset.

| Class | Training set (80%) | Validation set (20%) | Total samples |
|---|---|---|---|
| MEL | 890 | 223 | 1,113 |
| NV | 5,364 | 1,341 | 6,705 |
| BCC | 411 | 103 | 514 |
| AKIEC | 261 | 66 | 327 |
| BKL | 879 | 220 | 1,099 |
| DF | 92 | 23 | 115 |
| VASC | 113 | 29 | 142 |
| **Total** | **8,010** | **2,005** | **10,015** |

of the samples, while less prevalent conditions like "Dermatofibroma" (DF) comprise a mere 1.15%. This disproportionate representation can bias model training, leading to an overemphasis on the majority class (NV) and potentially overlooking crucial, albeit less frequent, diagnoses such as melanoma (MEL) [46]. Left unchecked, this imbalance can severely impact diagnostic accuracy, particularly for critical minority classes, which may be misclassified as noise or ignored altogether.

To mitigate this imbalance, we employed a random augmentation strategy applied specifically to the training set, focusing on balanced ratio control. Rather than collecting additional data, this approach leverages existing samples and applies a series of random transformations [47]. To ensure reproducibility while carefully preserving the essential diagnostic morphology of the lesions, a specific set of augmentations was applied to the training set images, each with a 50% probability of being executed. These transformations included geometric variations to simulate different image capture orientations, such as horizontal and vertical flips, and rotations within a conservative range of ±20 degrees. Additionally, to account for photometric variations that mimic slight changes in clinical lighting and focus, we adjusted image brightness by a factor of up to ±0.2, contrast by up to ±0.3, and applied a mild Gaussian blur with a sigma limit ranging from 0.1 to 1.0. Table 3 lists the specific parameters utilized for these transformations. The selection of these operations was specifically designed to simulate real-world imaging conditions and enhance the model's robustness. For instance, the application of Gaussian blur helps the model generalize to variations caused by skin texture or minor sensor noise, improving the clarity of lesion boundaries in images with artifacts or low contrast. Brightness and contrast adjustments mimic the inevitable variations in lighting conditions found in clinical settings, training the model to generalize across images captured with different levels of exposure. This is particularly useful for making crucial features like lesion borders and pigmentation patterns more prominent for the model to learn. Similarly, rotational transformations and flips address positional variability, ensuring the model's diagnostic capabilities are not dependent on the orientation of the lesion at the time of image capture.

These parameters were deliberately chosen to be conservative, introducing realistic diversity into the dataset without fundamentally altering the underlying pathological features. For instance, the limited rotation angle prevents extreme orientations that could distort key asymmetrical features. Crucially, a representative sample of the augmented images was visually inspected to confirm that lesion borders, texture, and color patterns remained clinically plausible and that no diagnostically relevant information was destroyed in the process. This careful, class-wise augmentation, guided by balanced ratio control, aims to amplify the representation of minority classes while carefully managing the overall class proportions relative to the dominant NV class. The resulting distribution of augmented training samples is visualized in Table 4. This balanced augmentation strategy enhances the training data's diversity and representativeness, promoting a more robust and reliable classification framework.

## Backbone models

This study employs three foundational deep learning architectures-DenseNet201, ResNet50V2, and ViT-as backbones for constructing six distinct fusion models. These architectures were selected for their demonstrated strengths in image

**Table 3**. Hyperparameters utilized for data augmentation.

| Augmentation Technique | Parameter Setting | Probability |
|---|---|---|
| *Geometric Transformations* | | |
| Rotation Range | ±20° | 0.5 |
| Horizontal Flip | True | 0.5 |
| Vertical Flip | True | 0.5 |
| *Photometric Transformations* | | |
| Brightness Adjustment | ±0.2 | 0.5 |
| Contrast Adjustment | ±0.3 | 0.5 |
| Gaussian Blur | Sigma ∈ [0.1, 1.0] | 0.5 |

**Table 4. Class distribution of the training dataset before and after the application of data augmentation.**

| Class | Training set (80%) | Augmented training set (80%) |
|-------|--------------------|------------------------------|
| MEL | 890 | 5,340 |
| NV | 5364 | 5,364 |
| BCC | 411 | 5,343 |
| AKIEC | 261 | 5,220 |
| BKL | 879 | 5,274 |
| DF | 92 | 5,336 |
| VASC | 113 | 5,311 |
| **Total** | **8,010** | **37,188** |

analysis tasks and their diverse representational capabilities. A brief overview of each model is provided below, with a more detailed exploration of the fusion strategies presented in the subsequent section.

1. Vision Transformer

The Vision Transformer (ViT) [48], emerges as a powerful architecture for image analysis. Departing from traditional convolutional networks, ViT leverages the transformer architecture, originally conceived for natural language processing. In our study, we employed ViT-B/16 architecture, the input image is decomposed into a grid of $16 \times 16$ pixel patches. These patches are then treated as individual tokens, analogous to words in a sentence. Each patch is subsequently flattened and linearly embedded, effectively transforming image data into a sequence of vectors. Positional embeddings are incorporated to retain spatial information, crucial for understanding image context.

This sequence of embedded patches is then fed into a standard transformer encoder. The encoder is composed of multiple layers of multi-head self-attention and feed-forward networks. The resulting architecture allows ViT model to capture long-range dependencies between image regions, facilitating a global understanding of the image content. The ability to model global context and complex feature interactions is important for skin lesion analysis, where subtle visual cues can indicate malignancy. ViT's inherent scalability and strong performance on large datasets make it an attractive option for automated skin cancer detection. Leveraging pre-trained on extensive datasets like ImageNet -which contains a wide range of object categories- ViT model demonstrates a strong understanding of visual features [48].

2. ResNet50V2

ResNet50V2 [49] is an enhanced iteration of the ResNet model family. ResNet architectures are renowned for their innovative use of residual learning, which addresses the vanishing gradient problem often encountered in deep networks. By employing shortcut connections that skip one or more layers, they ensure efficient gradient flow during backpropagation and enable the training of considerably deeper networks. What sets ResNet50V2 apart is its use of pre-activation of the weight layers within each residual block. Unlike earlier version, it places Batch Normalization and ReLU activation layers before the convolutional layers, as described in [49]. This adjustment stabilizes training, enhances gradient flow, and ultimately boosts generalization. Pre-trained on the ImageNet-1k dataset, ResNet50V2 comes equipped with a deep understanding of general image features. This pre-training is invaluable for domains like skin cancer image classification, where the model can apply its learned knowledge to detect subtle visual patterns related to malignancy in dermatological images.

3. DenseNet201

DenseNet201 [50], part of the Densely Connected Convolutional Network family, stands out as a highly efficient architecture for image classification. Its key innovation lies in the dense connections between layers. Unlike traditional networks

-where layers connect sequentially- DenseNet201 ensures every layer in a dense block is directly connected to all subsequent layers. This dense connectivity promotes feature reuse, improves information flow, and enhances the exploration of features, resulting in compact and effective models [51]. Each layer receives feature maps from all previous layers and passes its outputs forward, creating a network that learns highly discriminative features while mitigating the vanishing gradient problem. Additionally, the architecture's growth rate offers precise control over the network's complexity. Pre-trained on the ImageNet dataset, DenseNet201 is well-equipped with a diverse understanding of image features. This makes it suited for transfer learning in specialized fields like skin cancer image analysis, where recognizing subtle and complex patterns is critical.

## Fusion models

### 1. FM1

The first fusion model suggested and investigated in this work is a deep learning architecture that leverages the DenseNet201 and ResNet50V2 architectures in parallel, aiming to extract complementary feature representations from the input skin image, as shown in Fig 2. The model input is a color image of dimensions $224 \times 224 \times 3$, which undergoes processing through each backbone network separately.

Let $I \in \mathbb{R}^{224 \times 224 \times 3}$ denote the input image. The DenseNet201 network ($f_{DenseNet}$) and ResNet50V2 network ($f_{ResNet}$) act as feature extractors, where each model processes the input to generate a feature map:

$$F_{DenseNet} = f_{DenseNet}(I) \in \mathbb{R}^{7 \times 7 \times 1920} \tag{1}$$

$$F_{ResNet} = f_{ResNet}(I) \in \mathbb{R}^{7 \times 7 \times 2048} \tag{2}$$

Following feature extraction, each feature map undergoes further processing using a Separable Convolutional layer with Batch Normalization to enhance feature representation while reducing computational complexity. This processing is defined as:

$$F'_{DenseNet} = BN(DSC(F_{DenseNet}, 2048, 3 \times 3)) \tag{3}$$

$$F'_{ResNet} = BN(DSC(F_{ResNet}, 2048, 3 \times 3)) \tag{4}$$

where BN represents the Batch Normalization layer, and DSC represents the depthwise separable convolution operation with 2048 filters and a $3 \times 3$ kernel size.

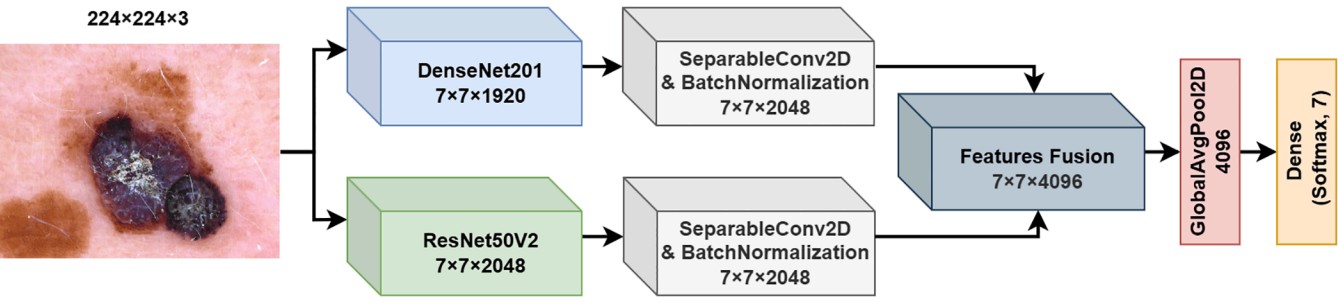

**Fig 2. Fusion model 1 (FM1) structure.**

The processed feature maps, $F'_{\text{DenseNet}}$ and $F'_{\text{ResNet}}$, are concatenated along the depth axis to create a unified feature representation:

$$F_{\text{fusion1}} = \text{Concat}(F'_{\text{DenseNet}}, F'_{\text{ResNet}}) \in \mathbb{R}^{7 \times 7 \times 4096} \tag{5}$$

This fusion strategy integrates the complementary features extracted by DenseNet201 and ResNet50V2, capturing a richer and more diverse set of patterns relevant to skin lesion classification.

The fused features $F_{\text{fusion1}}$ are then passed through a Global Average Pooling layer, which reduces the spatial dimensions by computing the average of each feature map, resulting in a 4096-dimensional vector:

$$F_{\text{pool}} = \text{GlobalAvgPool2D}(F_{\text{fusion1}}) \in \mathbb{R}^{4096} \tag{6}$$

Finally, the pooled features are fed into a fully connected (dense) layer with a softmax activation function for classification:

$$\hat{y} = \text{softmax}(W \cdot F_{\text{pool}} + b) \tag{7}$$

where $W$ and $b$ are the weights and bias of the dense layer, and $\hat{y} \in \mathbb{R}^7$ represents the predicted probability distribution over the seven skin lesion classes.

## 2. FM2

The second fusion model suggested and investigated in this work employs both DenseNet201 and ResNet50V2 backbones in parallel, with a novel channel-wise feature fusion strategy to enhance feature representation as shown in Fig 3. The same pre-processioning steps and initial feature extraction backbones are used similarly to FM1. The main difference between FM1 and FM2 is the way of fusion between the obtained features from Eqs (3) and (4). The feature maps $F'_{\text{DenseNet}}$ and $F'_{\text{ResNet}}$ are concatenated in a structured manner to create a fused feature map of size $14 \times 14 \times 2048$. The fusion process is conducted as follows:

1. Height Concatenation: The feature maps are concatenated along the height dimension:

$$F_{\text{concat\_height}} = \text{Concat}(F'_{\text{DenseNet}}, F'_{\text{ResNet}}) \in \mathbb{R}^{14 \times 7 \times 2048} \tag{8}$$

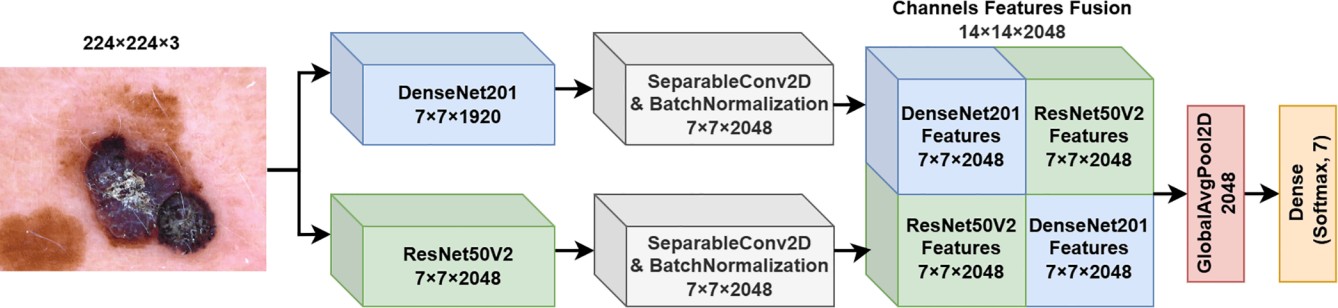

**Fig 3. Fusion model 2 (FM2) structure.**

2. Width Concatenation: The feature maps are concatenated along the width dimension:

$$F_{\text{concat\_width}} = \text{Concat}(F'_{\text{DenseNet}}, F'_{\text{ResNet}}) \in \mathbb{R}^{7 \times 14 \times 2048} \tag{9}$$

3. Zero Padding and Combination: The concatenated feature maps are padded with zeros to match the target shape $14 \times 14 \times 2048$, followed by combining them into a unified tensor:

$$F_{\text{fusion2}} = F_{\text{concat\_height\_padded}} + F_{\text{concat\_width\_padded}} \in \mathbb{R}^{14 \times 14 \times 2048} \tag{10}$$

This fusion strategy enhances spatial and channel-wise relationships between features extracted from the two backbone networks. The fused feature map $F_{\text{fusion2}}$ is then passed through a Global Average Pooling layer, which compresses the spatial dimensions, resulting in a feature vector of dimension 2048. Finally, the pooled feature vector is fed into a fully connected layer with a softmax activation function to produce the probability distribution over the seven skin lesion classes.

### 3. FM3

The third fusion model proposed in this work (FM3) extends the previous fusion strategies by incorporating a ViT alongside DenseNet201 and ResNet50V2 backbones, aiming to capture both convolutional and transformer-based feature representations. The architecture of FM3 is depicted in Fig 4. The model processes the input image through the three backbones in parallel and fuses their outputs through a novel spatial and channel-wise fusion strategy.

Let $I \in \mathbb{R}^{224 \times 224 \times 3}$ denote the input image. The DenseNet201 ($f_{\text{DenseNet}}$), ResNet50V2 ($f_{\text{ResNet}}$), and Vision Transformer ($f_{\text{ViT}}$) backbones act as feature extractors, processing the input image separately to generate feature maps. The DenseNet201 and ResNet50V2 models process the input image to produce feature maps $F_{\text{fusion2}}$ similarly to FM2. The ViT backbone $f_{\text{ViT}}$ processes the input image to produce a sequence of embeddings:

$$F_{\text{ViT}} = f_{\text{ViT}}(I) \in \mathbb{R}^{(1+N_p) \times D} \tag{11}$$

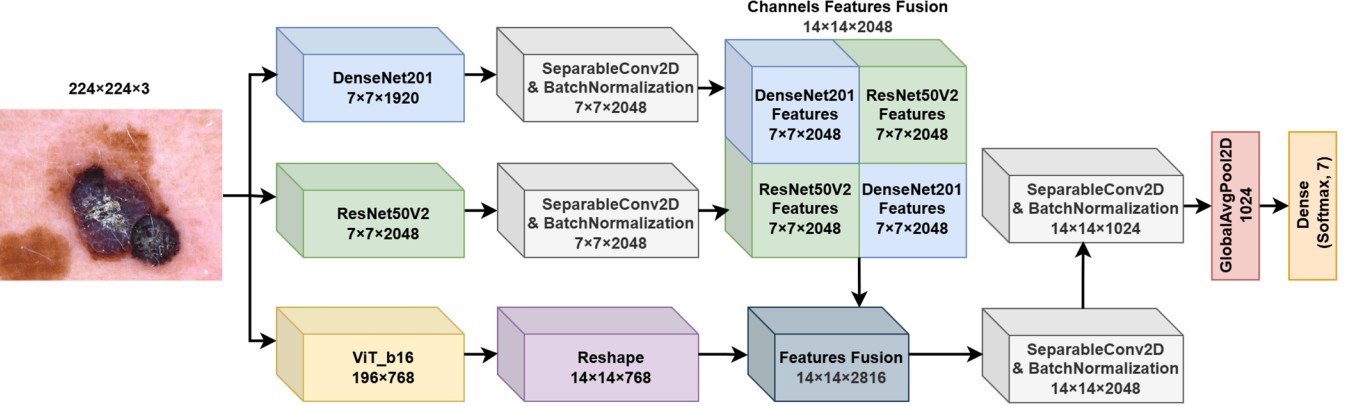

**Fig 4**. **Fusion model 3 (FM3) structure.**

where $N_p = 197$ is the number of image patches, and $D = 768$ is the embedding dimension. The first token corresponds to the class token, which is discarded in this fusion strategy. The remaining tokens are reshaped to form a feature map:

$$F'_{\text{ViT}} = \text{Reshape}(F_{\text{ViT}}) \tag{12}$$

This results in $F'_{\text{ViT}} \in \mathbb{R}^{14 \times 14 \times 768}$. The feature maps from ViT and the $F_{\text{fusion2}}$ are concatenated along the channel dimension to form a comprehensive feature representation:

$$F_{\text{combined}} = \text{Concat}(F'_{\text{ViT}}, F_{\text{fusion2}}, \text{axis} = 3) \in \mathbb{R}^{14 \times 14 \times 2816} \tag{13}$$

To further process the combined features, we apply a series of Separable Convolutional layers with Batch Normalization and ReLU activations. A Global Average Pooling layer is then applied to reduce the spatial dimensions. Finally, the pooled features are passed through the output layer with a softmax activation function for classification.

### 4. FM4

The fourth fusion model proposed in this work (FM4) introduces a novel fusion strategy by integrating the fused convolutional features from DenseNet201 and ResNet50V2 into a ViT architecture. Unlike FM3, where the ViT processes the input image separately, FM4 feeds the fused CNN features directly into the ViT model, enabling the transformer to capture global relationships within the fused feature space. The architecture of FM4 is depicted in Fig 5.

Similar to FM2, the image is processed through the DenseNet201 and ResNet50V2 backbones to extract feature maps, which are then fused into a single representation, $F_{\text{fusion2}}$. This fused feature map is subsequently reshaped and fed into the ViT model. To adapt $F_{\text{fusion2}}$ for input into the ViT, we reduce its channel dimension to match the ViT's embedding dimension ($D = 768$) using a Separable Convolutional layer:

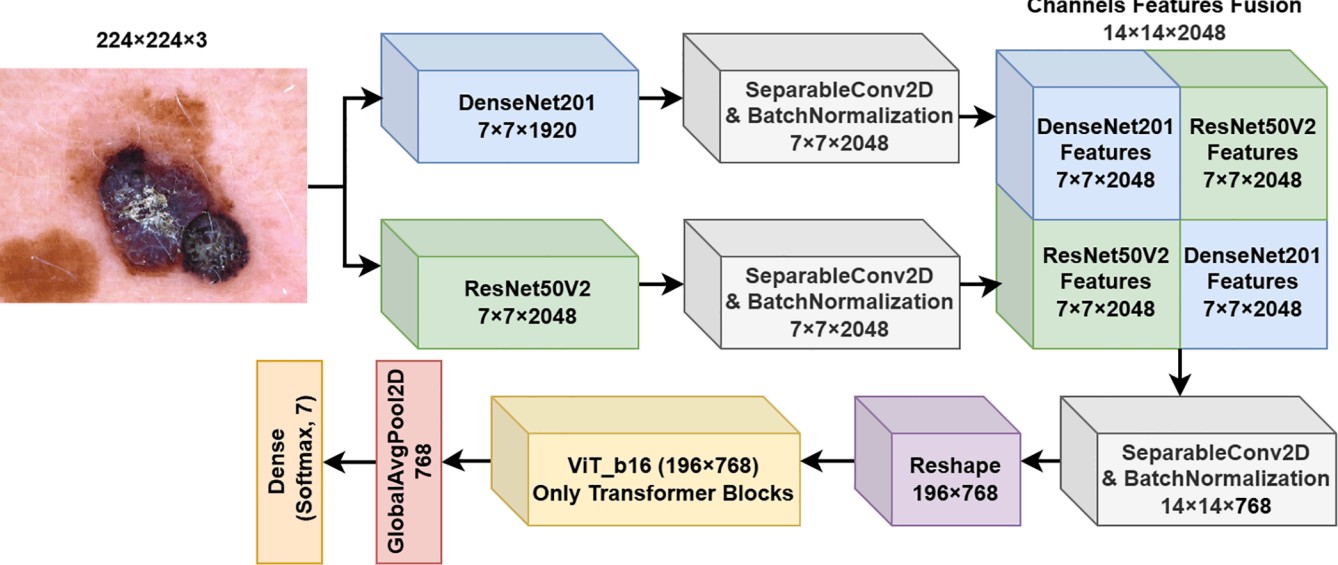

**Fig 5**. **Fusion model 4 (FM4) structure.**

$$F_{\text{embedded}} = \text{BN}(\text{DSC}(F_{\text{fusion2}}, 768, 3 \times 3)) \in \mathbb{R}^{14 \times 14 \times 768} \tag{14}$$

The fused and embedded feature map $F_{\text{embedded}}$ is reshaped into a sequence suitable for the ViT:

$$F_{\text{ViT\_input}} = \text{Reshape}(F_{\text{embedded}}, (N_p, D)) \in \mathbb{R}^{196 \times 768} \tag{15}$$

where $N_p = 14 \times 14 = 196$.

The sequence $F'_{\text{ViT\_input}}$ is then processed by the truncated ViT model encoder:

$$F_{\text{ViT\_output}} = f_{\text{ViT}}(F'_{\text{ViT\_input}}) \in \mathbb{R}^{197 \times 768} \tag{16}$$

A Global Average Pooling layer is then applied along the sequence dimension to aggregate the features. Finally, the pooled features are passed through a dense layer with a softmax activation function for classification.

## 5. FM5

The fifth fusion model proposed in this work (FM5) further extends previous fusion strategies by integrating upsampled feature maps from DenseNet201 and ResNet50V2 with the output from a ViT model. This approach allows FM5 to capture both fine-grained convolutional features and the global context of the ViT. The architecture of FM5 is shown in Fig 6.

Let $I \in \mathbb{R}^{224 \times 224 \times 3}$ denote the input image. The image is processed in parallel through the DenseNet201 ($f_{\text{DenseNet}}$) and ResNet50V2 ($f_{\text{ResNet}}$) backbones to extract feature maps, which are then passed through a Separable Convolutional layer, batch-normalized, and subsequently upsampled to obtain a consistent spatial resolution. The upsampling operation produces feature maps with dimensions 14×14×2048 for both DenseNet201 and ResNet50V2 outputs. At the same time, the ViT model processes the input image and generates a sequence of patch embeddings. A custom layer removes the class token, keeping only the patch embeddings, which are then reshaped to align with the spatial dimensions of the upsampled convolutional features. The upsampled feature maps $F'_{\text{DenseNet}}$, $F'_{\text{ResNet}}$, and $F'_{\text{ViT}}$ are concatenated along the channel dimension to form a comprehensive feature representation:

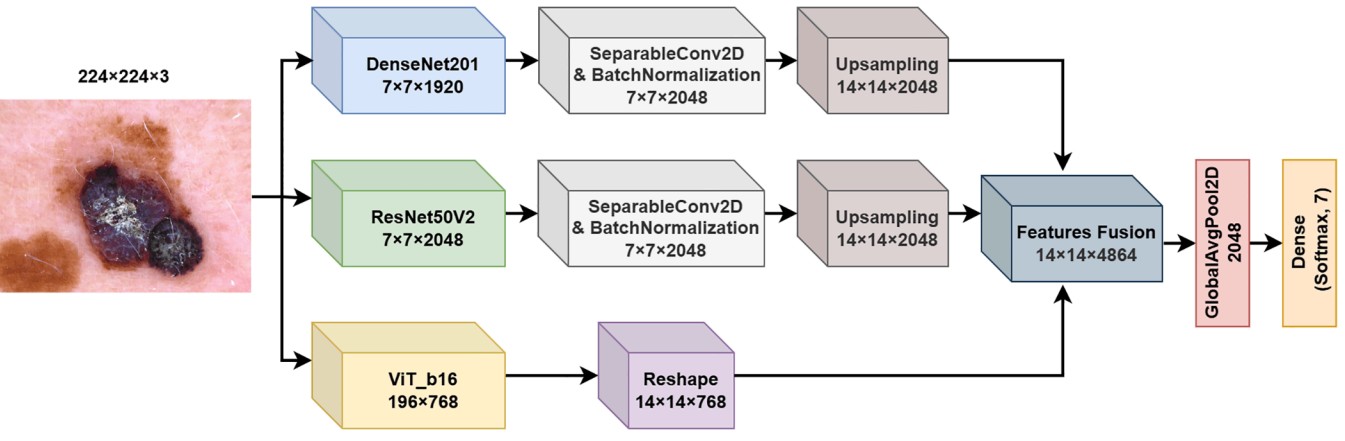

**Fig 6**. **Fusion model 5 (FM5) structure.**

$$F_{\text{combined}} = \text{Concat}(F'_{\text{DenseNet}}, F'_{\text{ResNet}}, F'_{\text{ViT}}, \text{axis} = 3) \in \mathbb{R}^{14 \times 14 \times 4864} \quad (17)$$

A Global Average Pooling layer is applied to the fused feature map $F_{\text{combined}}$ to reduce the spatial dimensions and produce a final feature vector. The pooled vector is then passed through a dense layer with a softmax activation function to output the predicted class probabilities.

### 6. FM6

The sixth fusion model proposed in this work (FM6) builds on previous architectures by combining feature maps from DenseNet201, ResNet50V2, and a ViT model, similar to FM5, but introduces additional dimensionality reduction on the ViT output through a Max Pooling layer. This fusion strategy aims to capture both the convolutional features and transformer-based representations while reducing computational complexity by pooling the ViT features. The architecture of FM6 is shown in Fig 7.

The image is processed in parallel through DenseNet201 ($f_{\text{DenseNet}}$) and ResNet50V2 ($f_{\text{ResNet}}$) backbones, producing feature maps $F'_{\text{DenseNet}}$ and $F'_{\text{ResNet}}$ through the operations defined in Eqs (3) and (4), followed by batch normalization. The resulting feature maps have dimensions $7 \times 7 \times 2048$. Simultaneously, the image is processed through a ViT model $f_{\text{ViT}}$, which outputs a sequence of patch embeddings. Using a custom layer, we discard the class token, and the remaining tokens are reshaped into a spatial feature map with dimensions $14 \times 14 \times 768$. To further reduce dimensionality and enhance computational efficiency, a Max Pooling layer is applied to $F'_{\text{ViT}}$, resulting in a downsampled feature map:

$$F''_{\text{ViT}} = \text{MaxPooling2D}(F'_{\text{ViT}}, (2, 2)) \in \mathbb{R}^{7 \times 7 \times 768} \quad (18)$$

The feature maps $F'_{\text{DenseNet}}$, $F'_{\text{ResNet}}$, and $F''_{\text{ViT}}$ are then concatenated along the channel dimension, forming a comprehensive feature representation:

$$F_{\text{combined}} = \text{Concat}(F'_{\text{DenseNet}}, F'_{\text{ResNet}}, F''_{\text{ViT}}, \text{axis} = 3) \in \mathbb{R}^{7 \times 7 \times 4864} \quad (19)$$

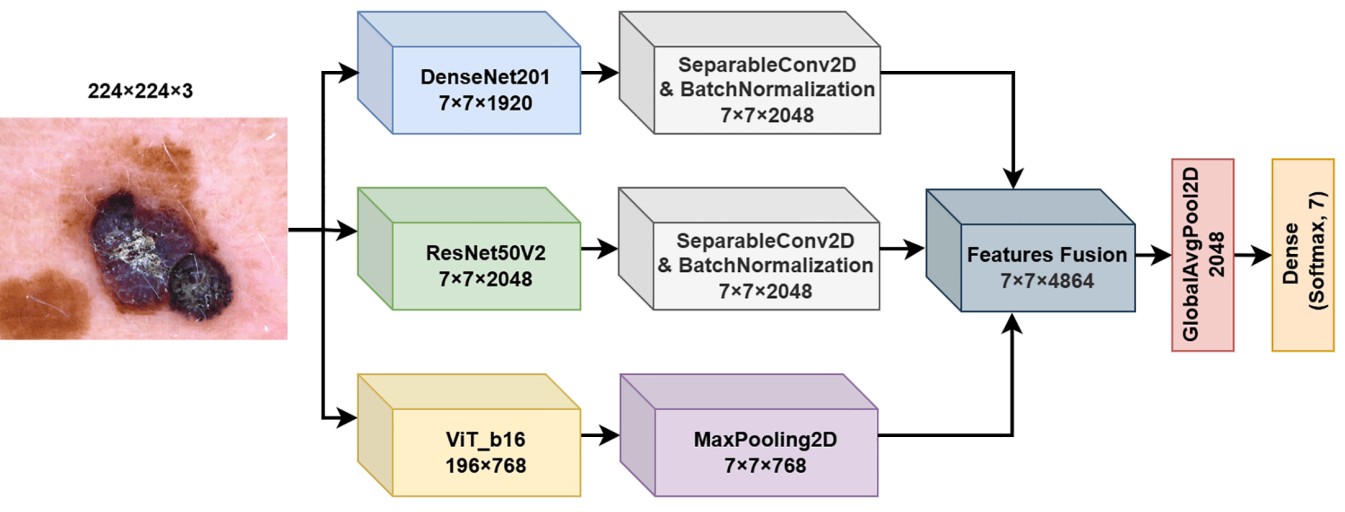

**Fig 7**. **Fusion model 6 (FM6) structure.**

A Global Average Pooling layer is applied to $F_{combined}$ to reduce the spatial dimensions and yield a final feature vector. This vector is fed into a dense layer with a softmax activation function and generates the predicted probability distribution across the skin lesion classes.

## Experiments and results

This section outlines the experimental setup and results of our skin lesion classification study. The experiments were carried out on a system running Ubuntu 22.04, 128GB of RAM, and an NVIDIA GeForce RTX 4090 GPU with 24GB of memory. TensorFlow 2.14 was used as the implementation framework. The following subsections provide an in-depth analysis of the proposed fusion models' performance and their ability to classify skin lesions. Additionally, the decision-making process is analyzed using Explainable AI techniques such as Grad-CAM and SHAP.

### Evaluation metrics

Model performance was rigorously evaluated using a suite of metrics to assess classification accuracy, efficiency, and the handling of false positives and negatives. A held-out validation set of 2,005 samples (20% of the total dataset, as detailed in Table 2) was used for this purpose [52]. The model's performance was evaluated using metrics designed to consider classification accuracy, efficiency, and the management of false positives and false negatives. For this evaluation, we used a validation set of 2,005 samples representing 20% of the total dataset as outlined in Table 2. The confusion matrix, representing the model's predictions in terms of true positives (TP), false positives (FP), true negatives (TN), and false negatives (FN), served as the foundation for our evaluation. From this confusion matrix, we derived several key classification metrics: accuracy, precision, recall, and F1 score. Accuracy, calculated as shown in Eq (20), quantifies the overall percentage of correctly classified skin lesions. This metric is particularly informative when class distributions are balanced within the validation set. Precision (also known as a positive predictive value, or PPV, calculated as in Eq (21) measures the accuracy of positive predictions. In the context of skin lesion diagnosis, high precision is crucial for minimizing misdiagnoses and ensuring appropriate treatment. Recall (also known as sensitivity or true positive rate, or TPR, calculated as in Eq (22) quantifies the model's ability to correctly identify all positive cases. High recall is paramount in medical diagnostics, particularly for conditions like melanoma, where minimizing false negatives is critical for patient outcomes. The F1 score in Eq (23), the harmonic mean of precision and recall, provides a balanced assessment of overall performance. For multi-class classification, we report both macro-averaged and weighted-averaged metrics. The macro average, sensitive to minority class performance, represents the unweighted mean across all classes, while the weighted average accounts for class prevalence by weighting the metric based on the number of samples in each class [52].

Receiver operating characteristic (ROC) curves and the associated area under the curve (AUC) provided further insights into model performance [52]. The ROC curve graphically depicts the trade-off between the true positive rate (TPR, Eq (22)) and the false positive rate (FPR) across different classification thresholds. The FPR represents the proportion of negative samples incorrectly classified as positive and is calculated as in Eq (24).

$$Accuracy = \frac{TP + TN}{TP + TN + FP + FN} \tag{20}$$

$$Precision\ (PPV) = \frac{TP}{TP + FP} \tag{21}$$

$$Recall\ (TPR) = \frac{TP}{TP + FN} \tag{22}$$

$$F1score = 2 \cdot \frac{Precision \cdot Sensitivity}{Precision + Sensitivity} \tag{23}$$

$$FPR = \frac{FP}{FP + TN} \tag{24}$$

## Results and discussion

The effectiveness of the six suggested feature fusion approaches for skin cancer multi-class classification is examined using the testing split of the HAM10000 dataset. Overall, the confusion matrices shown in Fig 8 emphasize the ability of different fusion techniques to capture the unique characteristics needed for accurate classification. In terms of overall accuracy, FM6 ranked as the best model, reaching 90.57% accuracy while FM3 and FM5 followed closely behind, recording accuracies of 90.42% and 90.27%, respectively. However, FM4 achieved the lowest accuracy (87.53%), indicating that its fusion strategy might not be as appropriate for skin cancer classification. Each model showed distinct strengths across lesion kinds when the class-wise performance was analyzed, indicating their differing abilities to manage the

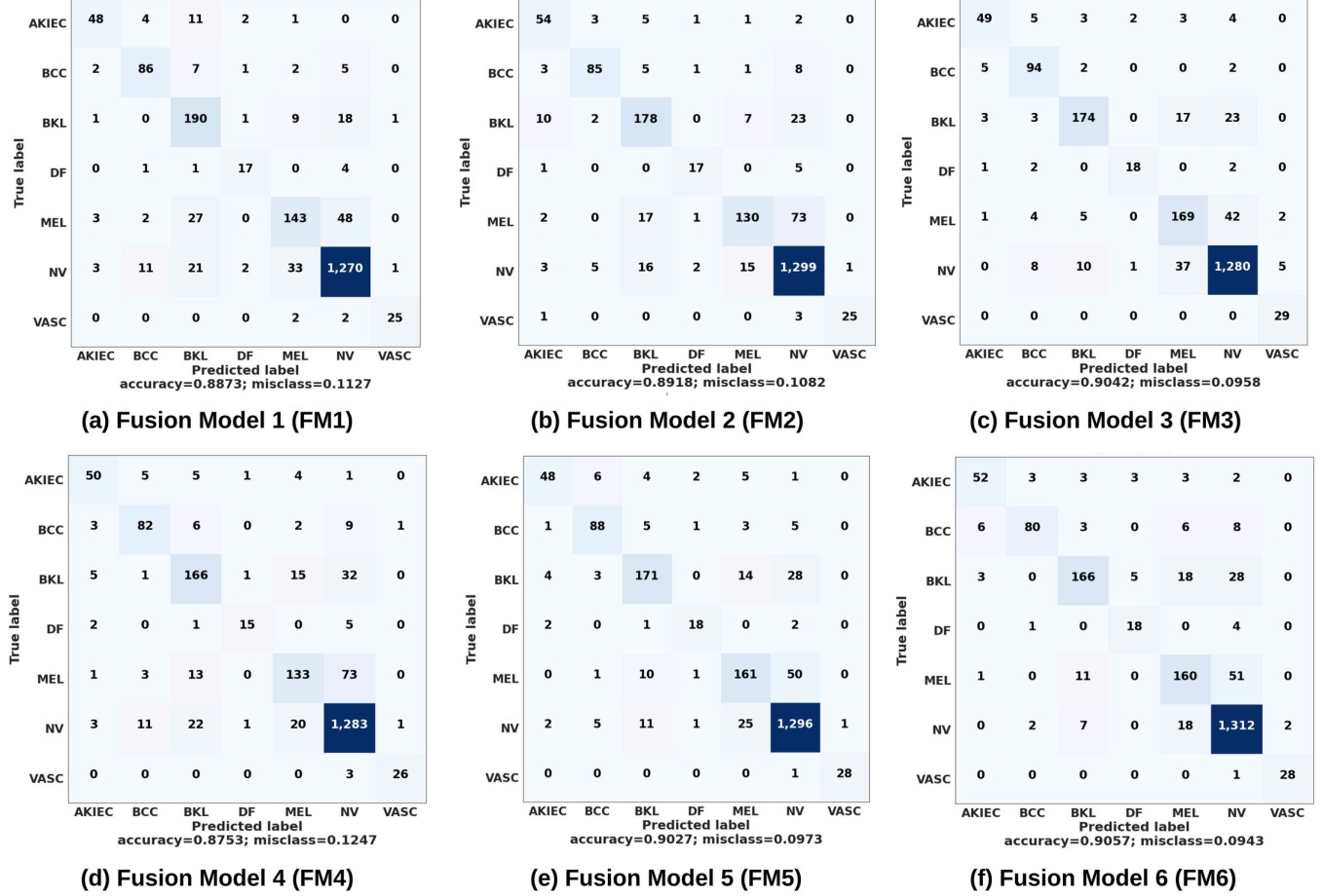

**Fig 8. Confusion matrices for the six suggested feature fusion ideas (FM1–FM6 models) on the HAM10000 dataset.**

dataset's complexity. With 52 and 51 true positives, respectively, FM6 showed good performance in detecting AKIEC and MEL, two crucial categories for the early identification of skin cancer. At 169 and 161, respectively, FM3 and FM5 demonstrated exceptional proficiency in detecting MEL instances. This is significant because, due to the increased risk of malignancy, precise melanoma identification is essential for medical treatments. On the other hand, DF, a less frequent lesion type in the dataset, presented difficulties for every model. Nonetheless, FM5 displayed comparatively fewer DF misclassifications, indicating that it may be more capable of capturing characteristics that set DF apart from other lesion types. In most classes, both FM1 and FM2 demonstrated moderate success, although they were not among the best models generally. Specifically, FM2 achieved marginally better results in distinguishing AKIEC and MEL than FM1. Notably, all models exhibited good classification accuracy for the NV class, which had the largest sample size and more distinguishing features. In the case of NV, FM6 delivered outstanding results achieving 1,312 accurate predictions. This performance underscores the model's ability to excel in identifying this relatively straightforward class. These findings emphasize the critical role of selecting the right fusion technique to optimize skin cancer classification.

The class-wise Receiver Operating Characteristic (ROC) curves and Area Under the Curve (AUC) values for each fusion model are presented in Fig 9. For the AKIEC and BCC classes, FM1, FM2, FM5, and FM6 achieved the highest AUC of 0.99. This indicates their exceptional ability to distinguish these classes from others. FM3 stood out for the BKL class and achieved an AUC of 0.98, highlighting its effectiveness in capturing the complex features of BKL lesions. For DF class, which is rare in the dataset, FM1, FM2, and FM5 models achieved the highest AUC of 0.99. This demonstrates their robustness in handling limited data. FM5 model excelled in melanoma (MEL) classification and it reached the highest AUC of 0.96. This is particularly significant given the malignancy risks associated with MEL, showcasing FM5's ability to enhance sensitivity for this critical class. In the case of NV, most of the proposed models achieved an AUC of 0.97. This consistency is crucial for clinical applications, as NV is one of the most common classes in the dataset. Lastly, almost all models showed perfect sensitivity and specificity for the VASC class achieving an AUC of 1.00 except FM4 which recorded 0.99. This result reflects the distinctiveness of VASC lesions, making them easier to differentiate. Overall, the FM5 and FM6 models emerged as top performers across most classes, showcasing their adaptability and strong ability to classify a wide range of skin lesion types.

Fig 10 illustrates the macro-average ROC curves for the six fusion models (FM1 to FM6) evaluated in a multi-class skin lesion classification task. All six models demonstrate excellent diagnostic performance with AUC values ranging from 0.97 to 0.98. FM1, FM2, FM5, and FM6 models lead the group with an AUC of 0.98 while FM3 and FM4 closely follow at 0.97. These results highlight the effectiveness of the proposed fusion strategies, particularly the depth-wise fusion methods employed by FM1, FM2, FM5, and FM6, which consistently outperform other approaches. The ROC curves for these models remain elevated and intersecting, indicating their ability to maintain high sensitivity (True Positive Rate) across a range of False Positive Rates. This consistency reflects strong and reliable classification performance. Among the models, FM5 and FM6 appear as the most stable and robust for automated skin lesion classification. The uniformly high AUC values across all models emphasize the potential of these frameworks for real-world clinical applications, providing a solid foundation for reliable AI-driven diagnostic tools.

The performance of the six fusion models (FM1–FM6) was further evaluated using classification metrics, including precision (P), recall (R), and F1 score in seven skin lesion classes, as detailed in Table 5. The FM1 approach effectively captures complementary properties from the DenseNet201 and ResNet50V2 backbones. However, the straightforward fusion strategy constrains the representation of complex interactions between features. The model scored a weighted average Precision, Recall, and F1 score of 0.88, signifying robust overall performance. However, for difficult classes such as AKIEC and MEL, the F1 scores were comparatively low at 0.78 and 0.73, respectively, underscoring the model's constrained capacity to successfully differentiate these classes. FM2, on the other hand, offers superior class-wise performance for categories like BCC and DF, with F1 scores rising to 0.85 and 0.75, respectively. This enhancement is due to the systematic spatial fusion approach that uses height and width fusion to greatly improve feature integration of the features obtained from the DenseNet201 and ResNet50V2 backbones. Although the weighted averages (P, R, and F1)

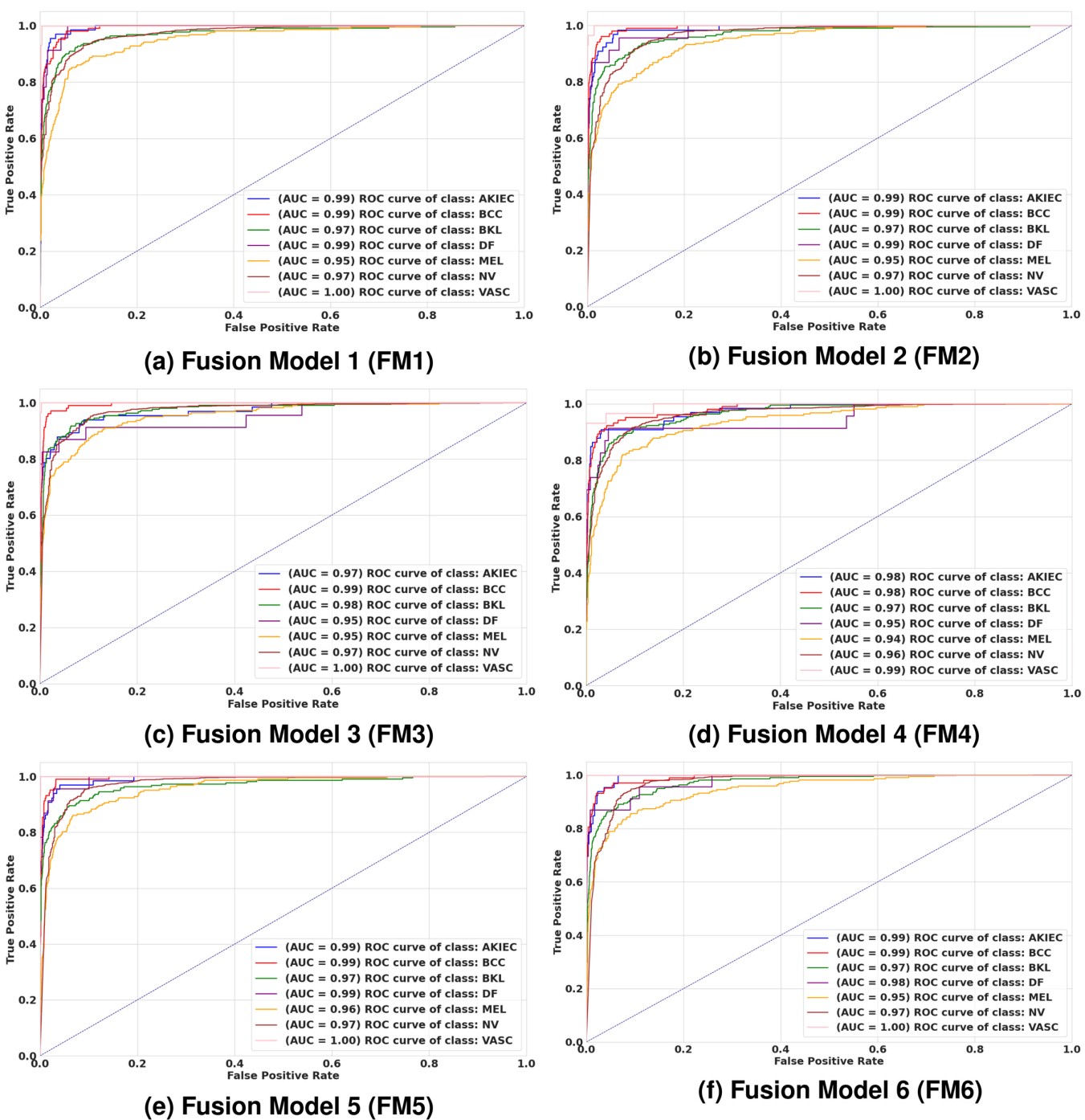

**Fig 9**. **Class-based ROC curves for the six suggested feature fusion ideas (FM1–FM6 models) on the HAM10000 dataset.**

are consistent with FM1 at 0.88, FM2 exhibits enhanced management of class imbalances, especially for well-represented categories. Nevertheless, the macro-average F1 score persists at 0.81, signifying minimal improvements for minority categories such as MEL.

**Fig 10. Average ROC curves for the six suggested feature fusion ideas (FM1–FM6 models) on the HAM10000 dataset.**

**Table 5. Comparison of classification metrics across different models on testing dataset.**

| Class | Fusion Model 1 (FM1) | | | Fusion Model 2 (FM2) | | | Fusion Model 3 (FM3) | | |
|---|---|---|---|---|---|---|---|---|---|
| | P | R | F1 | P | R | F1 | P | R | F1 |
| AKIEC | 0.84 | 0.72 | 0.78 | 0.72 | 0.81 | 0.77 | 0.83 | 0.74 | 0.78 |
| BCC | 0.82 | 0.83 | 0.83 | 0.89 | 0.82 | 0.85 | 0.81 | 0.91 | 0.85 |
| BKL | 0.73 | 0.86 | 0.79 | 0.80 | 0.80 | 0.80 | 0.89 | 0.79 | 0.84 |
| DF | 0.73 | 0.73 | 0.73 | 0.77 | 0.73 | 0.75 | 0.85 | 0.78 | 0.81 |
| MEL | 0.73 | 0.73 | 0.73 | 0.84 | 0.58 | 0.68 | 0.74 | 0.75 | 0.75 |
| NV | 0.94 | 0.94 | 0.94 | 0.91 | 0.96 | 0.94 | 0.94 | 0.95 | 0.95 |
| VASC | 0.92 | 0.86 | 0.89 | 0.96 | 0.86 | 0.90 | 0.80 | 1.00 | 0.89 |
| **Macro avg** | 0.82 | 0.80 | 0.81 | 0.84 | 0.80 | 0.81 | 0.84 | 0.85 | 0.84 |
| **Weighted avg** | 0.88 | 0.88 | 0.88 | 0.88 | 0.89 | 0.88 | 0.90 | 0.90 | 0.90 |
| Class | Fusion Model 4 (FM4) | | | Fusion Model 5 (FM5) | | | Fusion Model 6 (FM6) | | |
| | P | R | F1 | P | R | F1 | P | R | F1 |
| AKIEC | 0.78 | 0.75 | 0.76 | 0.84 | 0.72 | 0.78 | 0.83 | 0.78 | 0.81 |
| BCC | 0.80 | 0.79 | 0.80 | 0.85 | 0.85 | 0.85 | 0.93 | 0.77 | 0.84 |
| BKL | 0.77 | 0.75 | 0.76 | 0.84 | 0.77 | 0.81 | 0.87 | 0.75 | 0.80 |
| DF | 0.83 | 0.65 | 0.73 | 0.78 | 0.78 | 0.78 | 0.69 | 0.78 | 0.73 |
| MEL | 0.76 | 0.59 | 0.67 | 0.77 | 0.72 | 0.74 | 0.78 | 0.71 | 0.74 |
| NV | 0.91 | 0.95 | 0.93 | 0.93 | 0.96 | 0.95 | 0.93 | 0.97 | 0.95 |
| VASC | 0.92 | 0.89 | 0.91 | 0.96 | 0.96 | 0.96 | 0.93 | 0.96 | 0.94 |
| **Macro avg** | 0.82 | 0.77 | 0.79 | 0.85 | 0.82 | 0.84 | 0.85 | 0.82 | 0.84 |
| **Weighted avg** | 0.87 | 0.87 | 0.87 | 0.90 | 0.90 | 0.90 | 0.90 | 0.90 | 0.90 |

The addition of the ViT in FM3 enhances global contextual awareness in the fusion process. FM3 demonstrates substantial enhancements in difficult categories, including BKL (F1 = 0.84) and MEL (F1 = 0.75), by the integration of convolutional and transformer-based feature representations. The weighted average metrics increase to 0.90, signifying the model's proficiency in generalizing across all classes. Nonetheless, there is a marginal decline in performance for VASC (F1 = 0.89) relative to FM2 (F1 = 0.90), indicating that the incorporation of transformer-based features may intermittently diminish the emphasis on local features. FM4 adjusts the fusion technique by directly incorporating the fused CNN features from DenseNet201 and ResNet50V2 into the Vision Transformer. This method aims to utilize global relationships inside the integrated feature space at an early stage. However, this integration appears to be less effective, as demonstrated by slightly fragmented performance. The macro-average F1 score decreases to 0.79, while the weighted average metrics (Precision, Recall, F1) fall to 0.87. Despite FM4's strong performance in classes such as NV (F1 = 0.93), it encounters difficulties in others, such as MEL and DF, where F1 scores decrease to 0.67 and 0.73, respectively.

FM5 utilizes the upsampling of CNN features and concurrently integrates them with the output from the ViT. This method allows the model to efficiently incorporate both deep local details and global context. Consequently, FM5 attains balanced performance in every class, achieving a weighted average metrics of 0.90 and a macro-average F1 score of 0.84. Notably, FM5 performs well in underrepresented classes like MEL (F1 = 0.74) and BKL (F1 = 0.81). Building on FM5's success, FM6 uses a MaxPooling layer to add another dimensionality reduction step for the Vision Transformer output. This enhancement decreases computational complexity while preserving critical features, making FM6 a computationally efficient substitute for FM5. The macro-average F1 score of 0.84 and the weighted averages (P, R, F1) of 0.90 for FM6's performance indicators are comparable to those of FM5. The fusion model FM6 shows a slight drop in performance for specific classes like BCC, BKL, and DF but compensates with significant improvements in others such as AKIEC. Despite these fluctuations, it has robust overall results, excelling in key categories like NV (F1 = 0.95) and VASC (F1 = 0.94). These outcomes highlight FM6's ability to balance precision and recall effectively across diverse classes. Moreover, they demonstrate how the model leverages dimensionality reduction to enhance efficiency while retaining critical feature representations.

## Comparison with literature

This section benchmarks the proposed fusion models against state-of-the-art approaches for skin lesion classification on the HAM10000 dataset which was initially developed for the ISIC 2018 classification challenge. Table 6 presents a comparative performance overview, highlighting architectural differences, explainability integration, and key evaluation metrics. While several studies have addressed skin lesion classification using this dataset, many approach the problem as a binary classification or consider a reduced subset of classes, unlike the multi-class focus of this work. Moreover, even those studies achieving competitive performance often lack the crucial element of explainability, which is central to the robustness and interpretability of our proposed models. Alhudhaif et al. [27] investigate the impact of various data balancing strategies; their findings using augmentation, consistent with our methodology, provide a valuable benchmark. A particularly noteworthy case is the study by Jang and Park [31], reporting a remarkably high precision (P). However, this value stands in stark contrast to their other reported metrics (recall (R), F1 score (F1), and accuracy (Acc)), raising concerns about potential imbalances or limitations within their evaluation protocol. This discrepancy underscores the critical need for consistent performance across all metrics, a strength demonstrated by our proposed fusion models.

The proposed fusion models (FM1-FM6) exhibit consistently strong performance across all metrics, integrating both GradCAM and SHAP visualizations for comprehensive explainability. By leveraging diverse architectural components and fusion strategies, these models achieve improved accuracy while maintaining high precision, recall, and F1 scores. Specifically, Fusion Model 6 (FM6) attains the highest accuracy of 90.57%, demonstrating the potential of our approach for accurate, robust, and interpretable skin lesion classification.

**Table 6**. Comparison of the suggested fusion models with the existing approaches in the literature.

| Reference | | Model | | Number of classes | Statistical metrics | | | |
|-----------|---|-------|---|-------------------|---|---|---|---|
| Authors | Year | Model | XAI | | P (%) | R (%) | F1 (%) | Acc (%) |
| Aldwgeri and Abubacker [23] | 2019 | Ensemble Pre-trained models | No | 7 | 83.86 | 80.29 | 81.29 | 80.1 |
| Foahom Gouabou et al. [53] | 2021 | DDAG with VGG19 | No | 3 (melanoma, nevi, and seborrheic keratosis) | - | - | - | 76.6 |
| Ali et al. [32] | 2022 | EfficientNet B4 | No | 7 | 88 | 88 | 87 | 87.91 |
| Gouda et al. [54] | 2022 | InceptionV3 | No | 2 (malignant and benign) | - | - | - | 85.76 |
| Mridha et al. [25] | 2023 | CNN-based Framework | GradCAM, GradCAM++ | 7 | - | - | - | 81.24 |
| Alhudhaif et al. [27] | 2023 | CNN with Soft Attention Module (SAM) | No | 7 | 74.7 | 74.9 | 74.7 | 75.53 |
| Jang and Park [31] | 2024 | Dynamically Expandable Representation (DER) | GradCAM | 6, 8 | **91.8** | 80.8 | 84.7 | 80.88 |
| Yuan et al. [33] | 2024 | Self-feedback Threshold Focal Learning (STFL) | GradCAM | 7 | 74.26 | 77 | 74.62 | 77 |
| Mushtaq and Singh [55] | 2024 | Ensemble Visual Geometry Group-16 (EVGG-16) | No | 7 | 89 | 89 | 88 | 89 |
| Khan et al. [35] | 2024 | Multi-stage CNN pipeline with ELM | No | 7 | - | 86.98 | - | 87.02 |
| Setiawan & Soewito [36] | 2025 | CRCDKD with DenseNet | No | 7 | 83.27 | 84.85 | 83.39 | 89.41 |
| Suggested model 1 | 2025 | Fusion Model 1 (FM1) | GradCAM, SHAP | 7 | 88 | 88 | 88 | 88.73 |
| Suggested model 2 | 2025 | Fusion Model 2 (FM2) | GradCAM, SHAP | 7 | 88 | 89 | 88 | 89.18 |
| Suggested model 3 | 2025 | Fusion Model 3 (FM3) | GradCAM, SHAP | 7 | 90 | 90 | 90 | 90.42 |
| Suggested model 4 | 2025 | Fusion Model 4 (FM4) | GradCAM, SHAP | 7 | 87 | 87 | 87 | 87.53 |
| Suggested model 5 | 2025 | Fusion Model 5 (FM5) | GradCAM, SHAP | 7 | 90 | 90 | 90 | 90.27 |
| Suggested model 6 | 2025 | Fusion Model 6 (FM6) | GradCAM, SHAP | 7 | **90** | **90** | **90** | **90.57** |

## Grad-CAM heatmap

Grad-CAM (Gradient-weighted Class Activation Mapping) visualizations are essential for understanding the decision-making processes of deep learning models by emphasizing the spatial areas of the input picture that most significantly influence the model's predictions. These visualizations offer a comparative analysis of the efficacy with which each fusion model recognizes and concentrates on relevant lesion locations across various skin lesion categories [20,56,57]. Grad-CAM has been used to examine the ability of the six fusion approaches to detect and categorize skin lesions as shown in Fig 11. FM1 for example, exhibits limited spatial concentration in its Grad-CAM visuals. The activation maps generated by this model are broad and scattered, lacking the ability to reliably recognize crucial lesion regions. In challenging classes like NV, BKL, and VASC, FM1 emphasizes larger areas of the input images, most likely covering irrelevant background regions, which suggests weaker spatial discriminative capability. FM2 implements a structured spatial fusion strategy; nonetheless, its Grad-CAM maps are worse, indicating that the height and width concatenation method improves feature integration but fails to completely resolve the problem of spatial inconsistency. Despite having ViT components, FM3 and FM4 perform the worst in Grad-CAM visualizations. The activation maps for these models are disjointed and inconsistent, frequently inadequately emphasizing the critical lesion regions. In all classes, FM3 and FM4 concentrate on extraneous background regions or marginal locations, neglecting substantial amounts of the lesions. FM3 seeks to reconcile local and global characteristics by the amalgamation of convolutional and transformer-based representations; yet, the Grad-CAM results indicate that this integration generates noise and diminishes spatial concentration. Likewise, FM4's direct incorporation of fused CNN features into the ViT intensifies the problem, yielding activation maps that are more fragmented and less interpretable.

FM5 exhibits a significant enhancement in spatial accuracy as well as visualization in its Grad-CAM visuals. FM5 attains precise and localized activation maps by upsampling CNN features and merging them with ViT outputs.

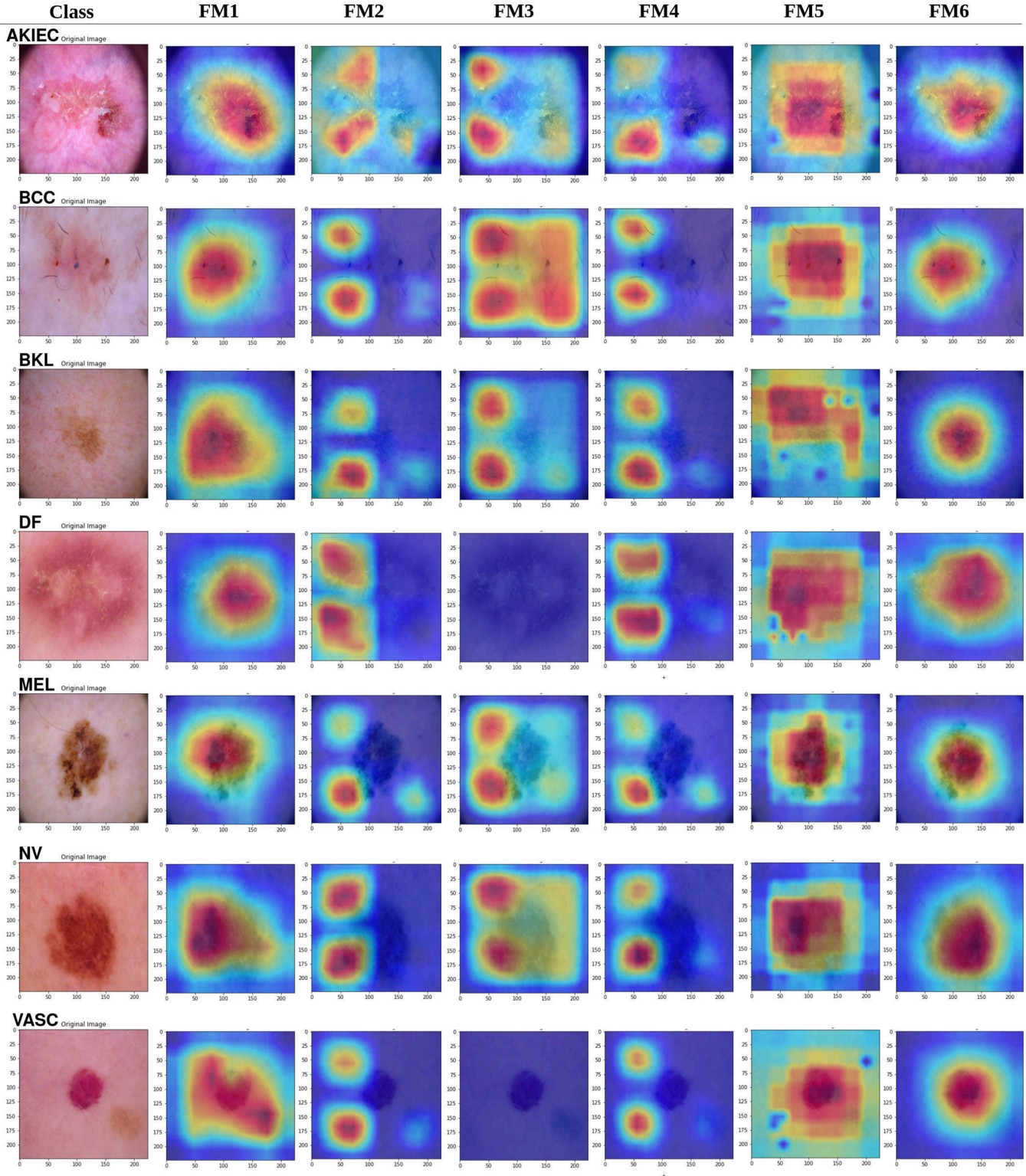

**Fig 11**. GradCAM results.

For demanding classes like AKIEC and MEL, FM5 highlights the lesion zones with remarkable accuracy, minimizing interference from surrounding areas. The improved spatial emphasis is particularly visible in Grad-CAM maps for categories such as NV and VASC, where the activations are both distinct and consistently cover the lesion margins. The results corroborate FM5's robust classification performance across all criteria, validating that the upsampling method significantly improves feature alignment and interoperability. The activation maps generated by FM6 are not only accurate and well-defined but also demonstrate less noise and fragmentation relative to all other models, including FM5. For instance, in the AKIEC and MEL categories, FM6 delineates the lesions with distinct margins and low background disturbance. This is a notable enhancement compared to FM3 and FM4, where fragmented activations were common. FM6's emphasis on lesion areas is apparent in well-represented classes such as NV and VASC, where its Grad-CAM maps consistently correspond with the lesion locations, hence strengthening its robust classification metrics (e.g., F1 = 0.95 for NV and F1 = 0.94 for VASC). The dimensionality reduction phase implemented in FM6 improves interpretability by refining the Vision Transformer output, enabling the model to concentrate on the most significant characteristics. To sum up, FM1 and FM2 exhibit diffuse activations, but FM3 and FM4 yield poor observations characterized by fragmented, noisy maps. FM5 enhances spatial accuracy with uniform, comprehensible maps across categories. FM6 enhances this through dimensionality reduction, attaining the clearest and least noisy activations, hence establishing it as the most reliable model for accurate skin lesion classification. Overall, the Grad-CAM representations of FM1, FM5, and FM6 show that these structures are capable of capturing high-level abstract properties and conceptual patterns. FM1 emphasizes fundamental components, but FM5 and FM6 are proficient in recognizing complicated patterns and significant areas crucial for accurate predictions. Conversely, FM2, FM3, and FM4 do not effectively capture relevant high-level features, frequently emphasizing irrelevant areas or insufficiently extracting essential patterns.

## Shapley additive explanations (SHAP)

SHAP (SHapley Additive exPlanations) is a game-theoretic method that provides feature-level interpretability by measuring the impact of individual pixel contributions on the model's output. While Grad-CAM highlights the general region of interest, SHAP offers a more granular view of the specific visual characteristics driving the model's decision. An analysis of SHAP visualizations in Fig 12 reveals a profound difference in the clinical relevance of the features learned by each model.

To ground this analysis in a clinical context, we can interpret the SHAP values through the lens of established dermatological heuristics, such as the widely used ABCD rule for melanoma detection, which assesses Asymmetry, Border irregularity, Color variegation, and Diameter. A truly interpretable model should base its decisions on these clinically recognized features. Our proposed models exhibit a wide spectrum of capabilities in this regard. The visualizations for models FM1 through FM4 show highly dispersed SHAP values, indicating a reliance on noisy or non-clinical features. For instance, the SHAP plot for FM4 applied to the MEL lesion is significantly fragmented, with feature importance scattered randomly across the lesion and the surrounding healthy skin. This demonstrates a failure to learn the structured, morphological signs of malignancy. In contrast, our top-performing model, FM6, demonstrates a remarkable ability to reason in a manner analogous to a trained dermatologist, as evidenced by the following case studies from Fig 12.

The melanoma sample, shown in the fifth row of Fig 12, is a textbook example of a high-risk lesion. It exhibits all the classic warning signs: it is highly Asymmetrical; its Border is poorly defined, irregular, and notched; and it displays significant Color variegation with multiple shades of dark brown and black. An ideal explainable model should prioritize these specific features. Our investigation reveals a clear hierarchy in this capability. FM5 and FM6 exhibit exceptional clinical alignment. Their SHAP plots concentrate high-importance (red) pixels precisely along the irregular portions of the border and within areas of heterogeneous pigmentation. Furthermore, additional clusters of high-importance pixels are located within the interior where the pigmentation is most heterogeneous. This output explicitly demonstrates that these models have learned to prioritize the very features—Border irregularity and Color variegation—that a clinician would identify as

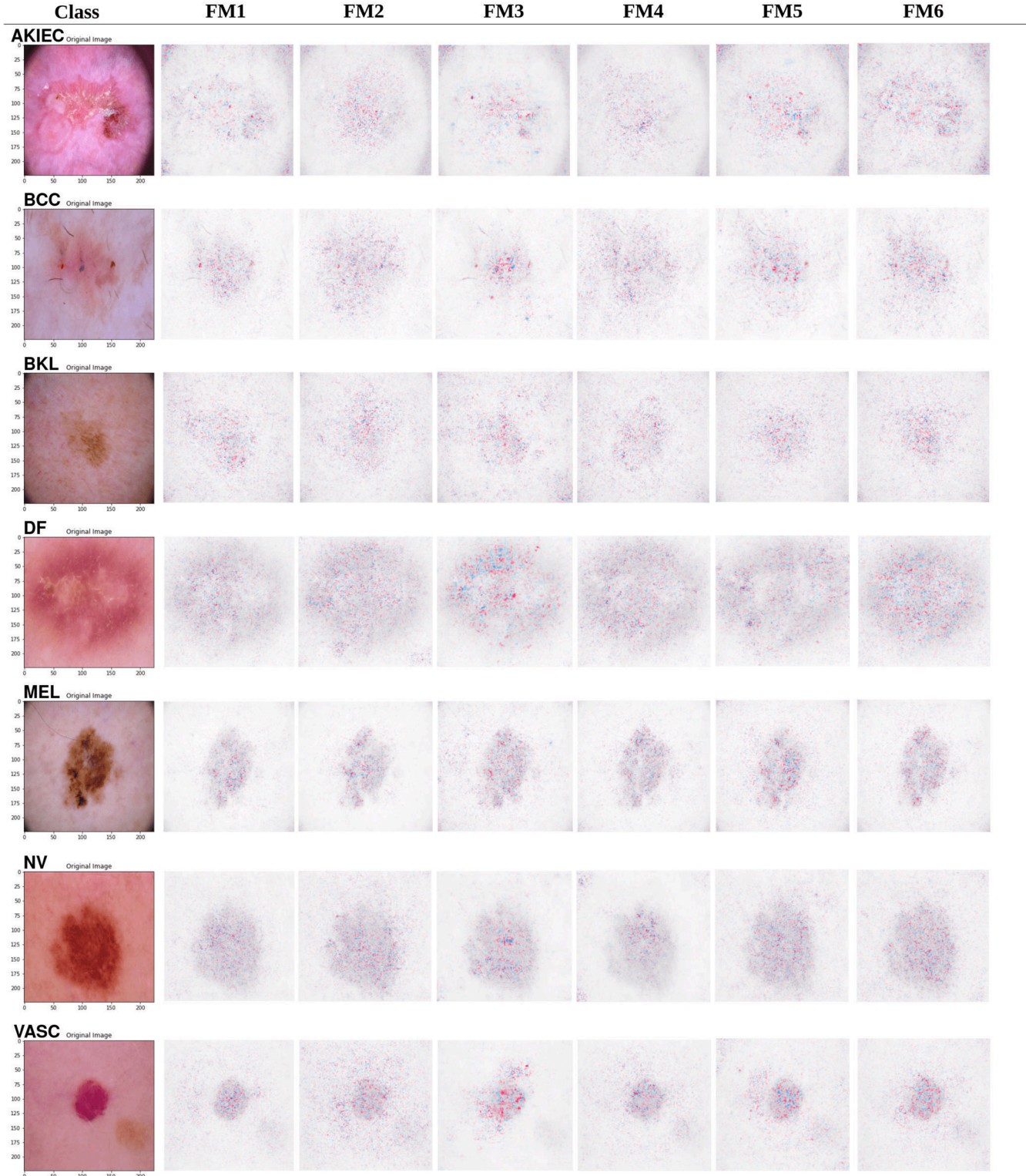

**Fig 12. SHAP values visualization.**

the strongest indicators of malignancy. FM1 and FM2, which rely on simpler CNN feature fusion, show a more diffuse focus. While their SHAP values are centered on the lesion, they fail to specifically isolate the critical border features, instead highlighting general color and texture. Their interpretation is plausible but lacks the clinical precision of FM5 and FM6. However, FM3 and FM4, which integrate ViT features more directly, produce the least interpretable results. Their SHAP plots are highly fragmented and noisy, scattering importance across the lesion, healthy skin, and irrelevant artifacts. They fundamentally fail to capture the morphological hallmarks of melanoma, making their predictions unreliable from an XAI perspective, even if the final classification is correct.

Basal Cell Carcinoma, while also malignant, often presents with different clinical features than melanoma. The BCC sample in the second row of Fig 12 shows a pinkish, pearlescent papule with subtle textural variations and a less defined border than a typical nevus. FM6 model, and to a similar extent FM5, again demonstrate sophisticated reasoning. They correctly shift their focus from the border (as in the melanoma case) to the lesion's internal surface morphology. Their SHAP plots highlight the subtle textural and chromatic variations within the papule, which are key diagnostic clues for this type of cancer. In contrast, FM1 through FM4 provide a much more generic interpretation. Their SHAP plots highlight the lesion as a non-specific "blob" of importance, failing to pinpoint the nuanced surface features that distinguish a BCC from other lesions. This suggests these fusion strategies produce models that are less capable of differentiating between the specific characteristics of various malignancies.

The power of an explainable model lies equally in its ability to correctly identify benign features. The Melanocytic Nevus (NV) in the sixth row is clinically benign, characterized by its relative symmetry, sharp and smooth Borders, and largely uniform Color. FM5 and FM6 models correctly reflect this benign nature. Their SHAP plots show importance distributed across the body of the lesion but, crucially, are quiet along the periphery. They have learned that a smooth, regular border is not an indicator of malignancy and therefore does not warrant high importance. For the texturally distinct Benign Keratosis (BKL, third row), they appropriately focus on its "stuck-on" surface texture. The other models, particularly FM3 and FM4, continue to produce noisy and clinically confusing visualizations. Their scattered SHAP values could incorrectly highlight random sections of a benign lesion's border, potentially creating a "false alarm" in an interpretability-driven review and eroding clinical trust.

In summary, this comparative analysis demonstrates that the choice of fusion architecture has profound implications for a model's clinical interpretability. While multiple models may achieve similar top-line accuracy, FM5 and FM6 stand apart in their ability to generate SHAP visualizations that are not only interpretable but are also clinically coherent and align with the diagnostic reasoning of a human expert. In contrast, the fusion strategies employed in FM1-FM4 result in models whose decision-making processes are more opaque and less reliable from an XAI standpoint. This investigation highlights that for a tool to be truly useful in a clinical setting, a synergy between predictive accuracy and robust, clinically-grounded interpretability is paramount.

## Diagnostic error analysis and interpretability consistency

To assess the clinical reliability of the FM6 architecture, a granular analysis of misclassified samples was conducted. This evaluation utilized explainable AI visualizations to identify the specific features contributing to erroneous predictions. Two primary failure modes were identified. Fig 13, first row, presents a false positive case where a benign Melanocytic Nevus was misclassified as Melanoma. The dermoscopic image contains air bubble artifacts resulting from the immersion fluid. The Grad-CAM heatmap exhibits peak spatial activation directly towards these bubbles. Concurrently, the SHAP visualization assigns positive feature importance, visualized as the presence of clusters of red pixels, specifically to the bubble edges. This confirms that the model incorrectly weighted these artifactual contours as malignant diagnostic features, revealing a sensitivity to non-biological noise. Fig 13, second row, illustrates a false negative case involving a Melanoma misclassified as a Melanocytic Nevus. Morphologically, this lesion exhibits low contrast and lacks a distinct border definition. The corresponding attention maps differ significantly from correctly classified samples. The Grad-CAM

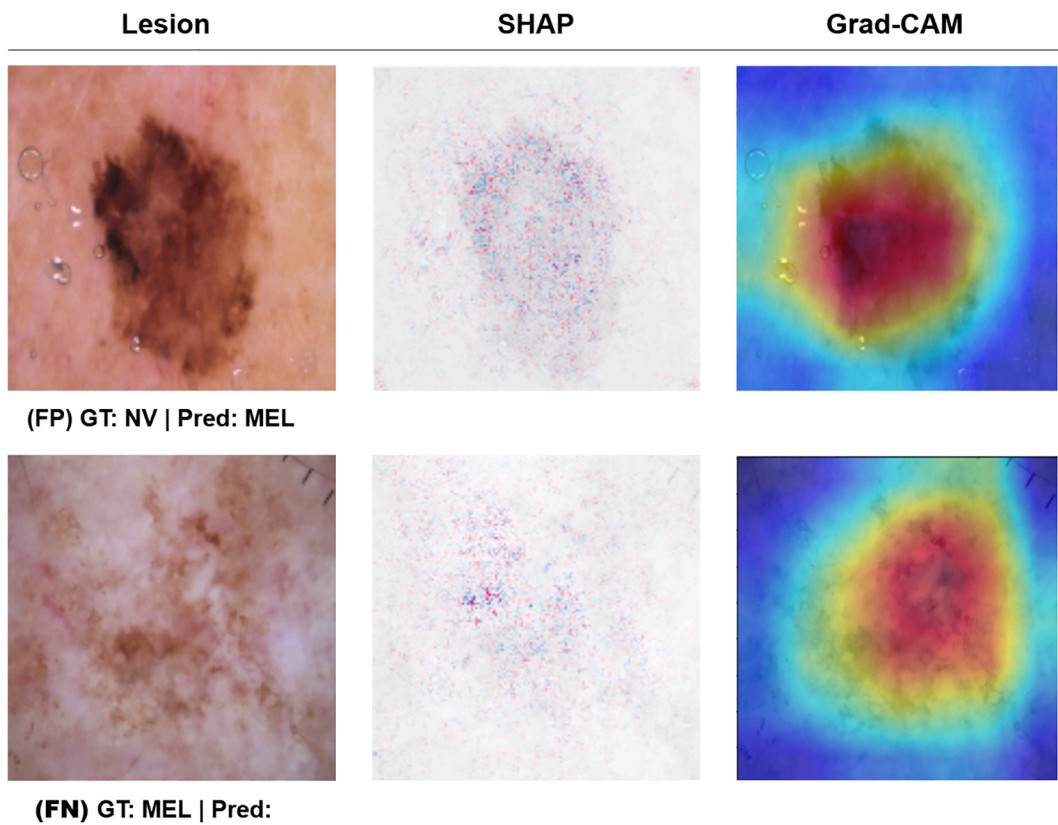

**Lesion** · **SHAP** · **Grad-CAM**

(FP) GT: NV | Pred: MEL

**(FN)** GT: MEL | Pred: NV

**Fig 13**. Diagnostic error analysis of the FM6 model comparing false positives and false negatives.

heatmap is diffuse and fails to localize the lesion boundary, while SHAP values are scattered. The inability to extract discriminative features from the indistinct border resulted in a default prediction of the majority class. A systematic comparison of the two interpretability frameworks reveals a correlation between prediction accuracy and XAI consistency. In correctly classified samples, Grad-CAM and SHAP predominantly highlighted identical regions. Conversely, erroneous predictions frequently exhibited divergence between spatial attention and feature attribution. This discordance suggests that inconsistency between XAI modalities may serve as a potential indicator of low diagnostic confidence in clinical workflows.

## Integration into clinical workflow

While the diagnostic accuracy and explainability of our proposed fusion models are promising, their true value lies in their potential for seamless integration into clinical dermatology workflows. A primary application would be a decision-support workflow during patient examinations. In this scenario, a dermatologist would capture a dermoscopic image of a suspicious lesion, which our model (ideally the top-performing FM6) would process in near-real time. The system would then present the clinician with a comprehensive output, comprising a primary diagnostic prediction with a corresponding confidence score, a ranked list of differential diagnoses, and the crucial explainable AI (XAI) visualizations. These visualizations would include a Grad-CAM heatmap overlaid on the original image to highlight the specific regions the model found most indicative of its prediction, alongside a SHAP visualization to provide insight into which visual features, such as irregular borders or specific color patterns, most heavily influenced the outcome.

The significance of this dual XAI output cannot be overstated, as it transforms the model from a "black box" into an interactive "second opinion." The Grad-CAM can draw the clinician's attention to subtle morphological features they may have overlooked, while the SHAP analysis helps build trust by providing a clear basis for the model's reasoning. The final diagnostic decision and patient management plan would, unequivocally, remain the purview of the human expert, who can now make a more informed decision by synthesizing their own expertise with the AI's quantitative analysis. A secondary, yet equally impactful, workflow is in teledermatology and triage. In high-volume settings, images captured by primary care physicians or nurses could be pre-screened by our system. Lesions flagged with a high probability of malignancy could then be prioritized for urgent review by a specialist, thereby optimizing resource allocation and potentially reducing diagnostic delays for critical cases. In this context, the model serves as an efficient tool to stratify risk and streamline the patient referral process. However, the transition from a research prototype to a deployed clinical tool involves navigating significant translational barriers. A primary challenge is device heterogeneity. Clinical settings employ a vast array of dermoscopes and camera systems which produce variations in color calibration and resolution. These variations can degrade model performance if the system is not sufficiently robust. Consequently, future development must focus on domain adaptation techniques to ensure consistent results across varying hardware. Regulatory approval also presents a critical step. To be certified as a Software as a Medical Device (SaMD), the system must demonstrate not only accuracy but also safety and generalizability across diverse patient demographics. This aligns with our focus on Explainable AI, as regulatory bodies increasingly demand transparency in algorithmic decision-making. Furthermore, ethical implications regarding accountability and automation bias must be managed. There is a risk that clinicians might over-rely on the AI output. Therefore, the presented XAI visualizations serve as a crucial check to ensure that the final diagnostic responsibility remains with the human expert.

## Conclusion and future works

This research presented an Explainable AI fusion model that integrates Vision Transformers (ViTs) and pre-trained Convolutional Neural Networks (CNNs) to improve the accuracy and reliability of automated melanoma diagnosis. The proposed method employs six fusion algorithms to efficiently merge complimentary feature representations from ViTs and CNNs, tackling issues related to complicated lesion patterns and variable quality of images in dermoscopic data. The top-performing fusion model attained a weighted average Precision, Recall, and F1 score of 90%, exhibiting strong performance across all classes, including difficult cases like melanoma and basal cell carcinoma. The study concluded that fusion along the depth axis is more effective than other axes, as it facilitates superior integration of complimentary feature representations from various backbone architectures. Depth-wise fusion enables the integration of hierarchical features derived from CNNs with global features calculated by the ViT, yielding enhanced and more robust feature maps. Moreover, the parallel integration of ViT with CNN backbones demonstrates greater potential than traditional methods, as it effectively integrates both local spatial features and global contextual features concurrently. CNNs proficiently capture complex patterns like edges and textures, whereas ViT enhances the model's ability to represent long-range dependencies and relationships within the image, hence properly managing complicated and variable lesion structures. Furthermore, the integration of XAI methodologies, including Grad-CAM and SHAP values, improves the model's reliability and transparency. Grad-CAM visuals emphasize the spatial regions that affect the model's predictions, whereas SHAP values offer a feature-level comprehension of the contributions influencing those predictions. These approaches show significant coherence with clinical intuition, especially in recognizing critical lesion regions and characteristics, including asymmetry, irregular boundaries, and color variations. The use of XAI guarantees that the model attains high accuracy while maintaining confidence among medical professionals through its delivery of interpretable insights into its decision-making process.

Despite these advances, the study recognizes several limits. The concurrent integration of ViT and CNNs has demonstrated efficacy; nonetheless, the computational burden persists as a barrier. Investigating the real-time implementation

of the model in resource-limited environments may greatly enhance accessibility and early detection rates in marginalized people. Moreover, performing comprehensive clinical trials and usability studies with healthcare professionals is crucial to enhancing the model's interface, improving workflow integration, and evaluating its effect on diagnostic efficiency and patient outcomes. An additional promising area for investigation is the fusion of multi-modal data, including the synthesis of dermoscopic images with patient metadata (e.g., age, UV exposure history, and family skin cancer history) to enhance diagnostic precision. By exploring various research avenues, the proposed approach could be enhanced into a whole diagnostic framework that not only attains high accuracy but also promotes extensive implementation in clinical settings. Finally, to validate the clinical utility of the proposed framework, we plan to conduct external validation. This will involve multi-center, prospective trials to test the system in real-time clinical workflows. These studies are necessary to confirm that the diagnostic improvements observed in this retrospective analysis translate effectively to patient care environments.

## Author contributions

**Conceptualization:** Humam AbuAlkebash, Radhwan A.A. Saleh.

**Data curation:** Humam AbuAlkebash, Radhwan A.A. Saleh.

**Formal analysis:** Humam AbuAlkebash, Radhwan A.A. Saleh.

**Investigation:** Humam AbuAlkebash, Radhwan A.A. Saleh.

**Methodology:** Humam AbuAlkebash, Radhwan A.A. Saleh.

**Project administration:** H. Metin ERTUNÇ.

**Resources:** H. Metin ERTUNÇ.

**Software:** Humam AbuAlkebash, Radhwan A.A. Saleh.

**Supervision:** H. Metin ERTUNÇ.

**Validation:** Humam AbuAlkebash, Radhwan A.A. Saleh.

**Visualization:** Humam AbuAlkebash, Radhwan A.A. Saleh.

**Writing – original draft:** Humam AbuAlkebash, Radhwan A.A. Saleh.

**Writing – review & editing:** H. Metin ERTUNÇ.

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
