## [Decision Letter · Decision Letter 0]

4 Aug 2025

PONE-D-25-11481Explainable Depth-Wise and Channel-Wise Fusion Models for Multi-Class Skin Lesion ClassificationPLOS ONE

Dear Dr. Saleh,

Thank you for submitting your manuscript to PLOS ONE. After careful consideration, we feel that it has merit but does not fully meet PLOS ONE’s publication criteria as it currently stands. Therefore, we invite you to submit a revised version of the manuscript that addresses the points raised during the review process.

We look forward to receiving your revised manuscript.

Kind regards,

Lin Xu

Academic Editor

PLOS ONE

Journal Requirements:

2. Please note that PLOS ONE has specific guidelines on code sharing for submissions in which author-generated code underpins the findings in the manuscript. In these cases, we expect all author-generated code to be made available without restrictions upon publication of the work.

Please review our guidelines at https://journals.plos.org/plosone/s/materials-and-software-sharing#loc-sharing-code and ensure that your code is shared in a way that follows best practice and facilitates reproducibility and reuse.

“This work was supported by the National Research Foundation of Korea (NRF) grant funded by the Korean government (MSIT) (No. RS-2023-00256517).”

“This work was supported by the National Research Foundation of Korea (NRF) grant funded by the Korean government (MSIT) (No. RS-2023-00256517). The authors also acknowledge the support of the Scientific Research Projects (BAP, Project Number: 3681) and the Kocaeli University Software Technologies Research (STAR) Laboratory, which facilitated the completion of this study.”

“This work was supported by the National Research Foundation of Korea (NRF) grant funded by the Korean government (MSIT) (No. RS-2023-00256517).”

Reviewers' comments:

Reviewer's Responses to Questions

**Comments to the Author**

1. Is the manuscript technically sound, and do the data support the conclusions?

Reviewer #1: Yes

Reviewer #2: Yes

2. Has the statistical analysis been performed appropriately and rigorously?

Reviewer #1: Yes

Reviewer #2: Yes

3. Have the authors made all data underlying the findings in their manuscript fully available?

Reviewer #1: Yes

Reviewer #2: Yes

4. Is the manuscript presented in an intelligible fashion and written in standard English?

Reviewer #1: Yes

Reviewer #2: Yes

5. Review Comments to the Author

Reviewer #1: The study addresses a critical gap in skin cancer diagnostics by integrating XAI with deep learning models.

A few constructive suggestions about this paper:

1. No discussion on how the proposed models would integrate into dermatology workflows

2. The manuscript claims "random augmentation" but lacks specifics on transformation parameters. Critical details like whether augmentation preserved lesion morphology are missing.

3. Without linking SHAP values to dermatological features, interpretability claims are incomplete.

4. Specify whether images were preprocessed (e.g., color normalization).

5. why do the authors use HAM10000 instead of ISIC2023?

Reviewer #2: This paper focuses on developing and rigorously evaluating novel explainable AI (XAI) special and channel fusion models for multi-class skin lesion classification. The research is of great practical significance, with sufficient workload and convincing experiments. However, from the perspective of the paper, improvements and innovations should be strengthened in the abstract section, focusing on how to solve problems. Secondly, the paper should include more recent research findings for discussion, such as 2024-2025. If the author can solve the above problems, we think the paper is excellent.

6. PLOS authors have the option to publish the peer review history of their article (what does this mean?). If published, this will include your full peer review and any attached files.

Reviewer #1: No

Reviewer #2: No

---

## [Author Response · Author response to Decision Letter 1]

11 Sep 2025

► Response Letter for “PONE-D-25-11481”

September 07, 2025

Dear Academic Editor Lin Xu,

We always hope you are fine and doing well.

We thank you for the opportunity to revise and resubmit our manuscript "Explainable Depth-Wise and Channel-Wise Fusion Models for Multi-Class Skin Lesion Classification" (PONE-D-25-11481). We appreciate the constructive comments and suggestions from you and the reviewers, which have helped us improve the quality of our work. We have now thoroughly revised the manuscript and have carefully addressed each suggestion point-by-point below. For your convenience, all changes have been highlighted in the “Revised Manuscript with Track Changes” file.

Thanks!

Journal Requirements

1. 1. Please ensure that your manuscript meets PLOS ONE’s style requirements, including those for file naming. The PLOS ONE style templates can be found at

and

Response:

We thank the editorial office for this reminder. We have taken several steps to ensure the revised manuscript fully complies with PLOS ONE’s formatting and style guidelines.

Specifically, we have downloaded and utilized the official PLOS ONE LaTeX template from the journal’s website for the revised version. We have also carefully reviewed the provided PDF templates for the main body and title page to ensure all sections, including affiliations, headings, figure captions, figure files, and reference formatting, align with the journal’s requirements. Besides, we have processed all figure files for our manuscript using the PACE tool as requested.

Furthermore, we confirm that upon resubmission, all files (including the “Response to Reviewers”, “Revised Manuscript with Track Changes”, and “Manuscript” files) will be named according to the specified conventions. We are confident that the revised submission now meets all PLOS ONE style requirements.

2. Please note that PLOS ONE has specific guidelines on code sharing for submissions in which author-generated code underpins the findings in the manuscript. In these cases, we expect all author-generated code to be made available without restrictions upon publication of the work.

Please review our guidelines at

https://journals.plos.org/plosone/s/materials-and-software-sharing#loc-sharing-code

and ensure that your code is shared in a way that follows best practice and facilitates reproducibility and reuse.

Response:

• Data availability statement: Page [39], lines [933-935]

• GitHub at (https://github.com/Radhwan-Saleh/SkinCancerFusionModel)

Thank you for the guidance on PLOS ONE’s code sharing policy. We have prepared a public repository containing the author-generated code used to produce the findings presented in our manuscript. The repository includes the model implementations, training and evaluation scripts, a detailed README file with instructions for reproduction, a list of dependencies, and a permissive open-source (MIT) license.

Furthermore, we have revised the “Data Availability Statement”, which provides a direct link to the public repository.

Revised Data availability statement:

“The dataset used in this study is publicly available. All source code is available on GitHub at (https://github.com/Radhwan-Saleh/SkinCancerFusionModel).”

"This work was supported by the National Research Foundation of Korea (NRF) grant funded by the Korean government (MSIT) (No. RS-2023-00256517)."

Response:

Thank you for the instruction. We have an important update regarding our funding situation, the full details of which are provided in our cover letter. The funding originally declared is no longer available, and the manuscript has been revised accordingly. Our updated and final financial disclosure statements are now as follows:

• Financial Disclosure: The authors received no specific funding for this work.

As no specific funding was received for this study, the "Role of Funder" statement is not applicable. We have noted this in our cover letter for the editorial team.

"This work was supported by the National Research Foundation of Korea (NRF) grant funded by the Korean government (MSIT) (No. RS-2023-00256517). The authors also acknowledge the support of the Scientific Research Projects (BAP, Project Number: 3681) and the Kocaeli University Software Technologies Research (STAR) Laboratory, which facilitated the completion of this study."

"This work was supported by the National Research Foundation of Korea (NRF) grant funded by the Korean government (MSIT) (No. RS-2023-00256517)."

Response:

We confirm that we have fully complied with this requirement. All funding-related text has been removed from the manuscript. To ensure clarity and full adherence to journal policy, the entire Acknowledgments section has been removed in this revised version.

Response:

We acknowledge the journal’s policy regarding recommended citations. We confirm that no specific publications were recommended for citation.

Response:

We thank the editorial office for this important instruction. We have performed a thorough review of our reference list for accuracy and completeness. To specifically check for retractions, we cross-referenced our entire bibliography against the Retraction Watch database, using their public dataset generated on August 22, 2025, and can confirm that none of our cited works have been retracted.

Reviewer 1

The study addresses a critical gap in skin cancer diagnostics by integrating XAI with deep learning models. A few constructive suggestions about this paper.

Response:

Thank you for your insightful and constructive review. We appreciate your recognition of our work’s significance and have carefully addressed each of your comments in the revised manuscript.

In the following, we provide point-by-point responses and outline the corresponding revisions made to improve the clarity, rigor, and overall quality of the paper. We believe these changes strengthen the manuscript and more clearly highlight its contributions to medical image analysis.

1. No discussion on how the proposed models would integrate into dermatology workflows.

Response:

• Page [37], lines [861-888]

We thank you for this relevant point. We agree that discussing the practical clinical utility of our proposed models is essential to framing the impact of our work.

To address this, we have added a new, dedicated subsection titled "Integration into Clinical Workflow" This section now elaborates on how our system could be deployed as a computer-aided diagnosis (CADx) tool, detailing its potential roles in both direct clinical decision-support, leveraging the dual XAI outputs as an interpretable, and in teledermatology triage to help prioritize patient care.

2. The manuscript claims "random augmentation" but lacks specifics on transformation parameters. Critical details like whether augmentation preserved lesion morphology are missing.

Response:

• Pages [15-16], lines [331-356]

Thank you for this critical comment regarding the need for methodological transparency and reproducibility. We have now substantially revised the “Data augmentation” section to address this thoroughly.

In the revised section, we now specify the exact transformations and their corresponding parameters used in our study, including horizontal and vertical flips (each with 50% probability), rotations (within a range of ±20 degrees), brightness adjustments (up to ±0.2) and contrast adjustments (up to ±0.3), and Gaussian blur (with a sigma limit of 0.1 to 1.0).

Furthermore, we explicitly address the point about preserving lesion morphology. We now state that these conservative parameters were deliberately selected to mimic realistic clinical variations (e.g., in lighting and orientation) without distorting diagnostically critical features. Most importantly, we have added a statement confirming that we visually inspected the augmented samples to ensure their clinical validity and the integrity of the lesion characteristics.

3. Without linking SHAP values to dermatological features, interpretability claims are incomplete.

Response:

• Pages [34-37], lines [787-860]

Thank you for this superb point, which gets to the very core of meaningful explainability in a clinical context. To rectify this, we have significantly revised and expanded the “SHapley Additive exPlanations (SHAP)” subsection. The analysis is no longer just a summary of heatmap quality but is now a direct interpretation of the model’s reasoning in a clinical context.

The key enhancements in the revised section are:

• Clinical Framework: We now explicitly introduce the ABCD rule as a clinical framework to provide a common language for interpreting the model’s feature importance.

• Detailed Case Studies: We perform a series of in-depth case studies using examples from Figure 12. This includes a comparative analysis of how all our fusion models interpret.

• Comparative Analysis: The entire subsection is now framed as a comparative analysis of fusion strategies, discussing the trade-offs in interpretability that each architectural choice entails. This directly aligns with the investigative spirit of our manuscript.

4. Specify whether images were preprocessed (e.g., color normalization).

Response:

• Page [14], lines [305-314]

We thank the reviewer for raising this point, which highlights the need for a more detailed methodological description. While we had summarized the key preprocessing steps in the overview diagram of our framework (Figure 1), we acknowledge that a thorough explanation of these steps was not adequately detailed in the main body of the manuscript, which is essential for transparency and reproducibility.

To rectify this, we have now added several new paragraphs to the “Dataset and preprocessing” subsection. This text provides a comprehensive and explicit account of our entire preprocessing pipeline. The steps, now clearly detailed, include:

• Resizing: We specify that all images were uniformly resized from their original resolution to 224x224 pixels to meet the input requirements of the backbone models.

• Intensity Normalization: We clarify the exact method of color normalization used, which was to scale the pixel values of each channel from their original integer range to a [0.0, 1.0] floating-point range by dividing by 255.

5. Why do the authors use HAM10000 instead of ISIC2023?

Response:

Thank you for this question regarding dataset selection. While the ISIC 2023 challenge dataset is not available on the official ISIC Archive, our analysis of the other recent challenges (2020, 2021, 2022, and 2024) confirms that HAM10000 was the most appropriate choice for our study.

The primary reason is task alignment. Our research objective is specifically multi-class classification across seven distinct lesion types from a single image. The recent ISIC challenges, in contrast, have consistently focused on binary classification (e.g., malignant vs. benign) or different modalities entirely (e.g., multimodal data, longitudinal 3D imaging). These tasks are fundamentally different from our research question.

Consequently, HAM10000 is the most widely-used benchmark for the multi-class classification task we address, making it essential for comparing our results against other work. This choice was necessary to fairly benchmark our fusion models against the field’s leading methods and validate our findings.

Reviewer 2

This paper focuses on developing and rigorously evaluating novel explainable AI (XAI) special and channel fusion models for multi-class skin lesion classification. The research is of great practical significance, with sufficient workload and convincing experiments.

Response:

Thank you for your positive and we appreciate your recognition of the importance of our proposed framework for skin cancer detection and classification. We also value your constructive suggestions for further improvements. We have carefully considered each of your comments and have made revisions accordingly to enhance the clarity, depth, and rigor of our work. Below, we provide detailed responses to your suggestions and the corresponding updates made in the manuscript.

1. From the perspective of the paper, improvements and innovations should be strengthened in the abstract section, focusing on how to solve problems.

Response:

• Pages [1-2]

We thank the reviewer for this insightful feedback. Upon review, we recognized that while our abstract accurately described the components of our study, it did not effectively communicate the central innovation or the problem-solving narrative.

To address this, we have completely rewritten the abstract to be more focused and impactful. The new version has been restructured to tell a clearer story, as follows:

• It now explicitly frames the research around solving the dual challenge of achieving both high accuracy and clinical transparency in dermatological AI.

• It introduces our core innovation upfront: the systematic investigation of fusing the local feature expertise of CNNs with the global contextual understanding of Vision Transformers (ViTs) as our specific solution to this problem.

• The abstract now more clearly links our quantitative results (90% F1-score) to solving the accuracy problem and our XAI analysis (demonstrating alignment with clinical reasoning) to solving the transparency problem.

2. The paper should include more recent research findings for discussion, such as 2024-2025. If the author can solve the above problems, we think the paper is excellent.

Response:

Thank you for this suggestion. We have conducted a thorough literature search and have identified four highly relevant, state-of-the-art studies published in 2024 and 2025. We have now carefully integrated these recent findings into the manuscript to provide a more comprehensive and up-to-date discussion.

These significant updates have been made in two key sections of the paper:

1- In the "Related Works" Section, Pages [8-9, 10-11], lines [194-213] and lines [265-275].

2- In the "Comparison with literature" Section, Table 5, Page [32].

---

## [Decision Letter · Decision Letter 1]

1 Dec 2025

PONE-D-25-11481R1Explainable Depth-Wise and Channel-Wise Fusion Models for Multi-Class Skin Lesion ClassificationPLOS ONE

Dear Dr. Saleh,

Thank you for submitting your manuscript to PLOS ONE. After careful consideration, we feel that it has merit but does not fully meet PLOS ONE’s publication criteria as it currently stands. Therefore, we invite you to submit a revised version of the manuscript that addresses the points raised during the review process.

We look forward to receiving your revised manuscript.

Kind regards,

Lin Xu

Academic Editor

PLOS ONE

Journal Requirements:

Reviewer's Responses to Questions

**Comments to the Author**

1. If the authors have adequately addressed your comments raised in a previous round of review and you feel that this manuscript is now acceptable for publication, you may indicate that here to bypass the “Comments to the Author” section, enter your conflict of interest statement in the “Confidential to Editor” section, and submit your "Accept" recommendation.

Reviewer #1: All comments have been addressed

Reviewer #3: (No Response)

2. Is the manuscript technically sound, and do the data support the conclusions?

Reviewer #1: Yes

Reviewer #3: Partly

3. Has the statistical analysis been performed appropriately and rigorously?

Reviewer #1: Yes

Reviewer #3: N/A

4. Have the authors made all data underlying the findings in their manuscript fully available?

Reviewer #1: Yes

Reviewer #3: Yes

5. Is the manuscript presented in an intelligible fashion and written in standard English?

Reviewer #1: Yes

Reviewer #3: Yes

6. Review Comments to the Author

Reviewer #1: Dear authors, after careful review, I have no additional concerns regarding your submission at this time.

Reviewer #3: The paper presents a comprehensive study on explainable multi-class skin lesion classification by designing six fusion-based deep learning models that combine the local feature extraction strength of CNNs with the global contextual capabilities of Vision Transformers (ViTs). Using the challenging 7-class HAM10000 dataset, the authors rigorously evaluate depth-wise and channel-wise fusion strategies and achieve a strong weighted Precision, Recall, and F1 of 90%. A key contribution of this work is its emphasis on clinical interpretability, where the authors employ Grad-CAM and SHAP to show that the best-performing fusion models focus on medically meaningful attributes such as border irregularity and color variegation—aligning with dermatological reasoning. Overall, the paper offers an accurate, transparent, and clinically aligned AI framework, highlighting how fusion architecture choices directly influence both performance and explainability in automated skin cancer diagnosis

Comments:

1. Transparency and Reproducibility

The authors should provide full open-source code, pretrained models, and data preprocessing scripts in a publicly accessible repository. Including reproducible

Although code is included, it's recommended that a Jupyter notebooks for training, evaluation, and ablation would demonstrate commitment to transparency.

2. Comprehensive Dataset and Method Details

Expand dataset descriptions with detailed information about patient/sample demographics, imaging modalities, annotation standards, and multi-center data harmonization. Include explicit preprocessing and augmentation details to improve reproducibility.

3. Robust Experimental Design and Statistical Rigor

Implement stratified k-fold cross-validation or repeated random splits with detailed fold-wise metrics. Present statistical significance testing (e.g., paired t-test, Wilcoxon) for key results and detailed per-class performance breakdowns.

4. Expanded Explainability Approaches

Integrate additional explainability frameworks (e.g., Grad-CAM, LIME, SHAP) with relevant qualitative and quantitative analyses. Include case studies explaining both successes and errors to support clinical interpretability.

5. Ablation and Component Impact Analysis

Provide detailed ablations for model components, data preprocessing steps, and optimization algorithms. Report both quantitative metrics and qualitative results indicating the contribution of each component.

6. Broader Benchmarking

Benchmark against state-of-the-art deep learning methods relevant to the problem, including attention-based, transformer, and hybrid architectures. Include efficiency comparisons relevant for clinical use.

7. Handling Imbalanced Data and Rare Cases

Discuss strategies employed for imbalanced data or rare conditions, reporting per-class metrics, and experiments with weighted losses or data augmentation.

8. Clinical Integration and Translation

Elaborate on considerations for real-world deployment in clinical settings, regulatory challenges, device heterogeneity, and ethical implications. Propose plans for external, multicenter, or prospective validation.

9. Enhanced Literature Integration

Expand literature review to include recent advances and foundational works in biomedical AI, especially explicability, federated learning, and clinical translation, citing:

https://www.sciencedirect.com/science/article/pii/S1476927125000283

https://link.springer.com/article/10.1007/s11831-025-10255-2

https://www.sciencedirect.com/science/article/pii/S1476927125001604?via%3Dihub

https://www.nature.com/articles/s41598-025-12602-6

https://pubmed.ncbi.nlm.nih.gov/40751377/

https://www.nature.com/articles/s41598-025-11574-x

https://link.springer.com/article/10.1007/s11831-025-10315-7

https://www.nature.com/articles/s41598-025-14333-0

https://link.springer.com/article/10.1007/s11831-025-10376-8?utm_source=rct_congratemailt&utm_medium=email&utm_campaign=nonoa_20251002&utm_content=10.1007%2Fs11831-025-10376-8

https://link.springer.com/article/10.1007/s11831-025-10379-5

https://link.springer.com/article/10.1007/s11831-025-10414-5

https://link.springer.com/article/10.1007/s11831-025-10411-8

https://www.sciencedirect.com/science/article/pii/S1746809425013631?r

https://journals.tubitak.gov.tr/biology/vol49/iss5/9/

7. PLOS authors have the option to publish the peer review history of their article (what does this mean?). If published, this will include your full peer review and any attached files.

Reviewer #1: No

Reviewer #3: No

---

## [Author Response · Author response to Decision Letter 2]

19 Dec 2025

Journal Requirements:

Response:

We thank the Editor for this guidance. In accordance with the Journal Requirements, we have carefully evaluated the list of publications suggested during the review process. We applied a systematic evaluation strategy to determine their scientific relevance to our study's scope, specifically prioritizing works that address hybrid architectures and explainable AI in medical imaging. Consequently, we have incorporated only those references that directly enhance the quality and contextual depth of our manuscript, ensuring that all citations are scientifically justified.

Reviewer #1:

Dear authors, after careful review, I have no additional concerns regarding your submission at this time.

Response:

Thank you for your insightful and constructive review.

Reviewer #3:

Thank you for your positive and we appreciate your recognition of the importance of our proposed framework for skin cancer classification. We also value your constructive suggestions for further improvements. We have carefully considered each of your comments and have made revisions accordingly to enhance the clarity, depth, and rigor of our work. Below, we provide detailed responses to your suggestions and the corresponding updates made in the manuscript.

1. Transparency and Reproducibility: The authors should provide full open-source code, pretrained models, and data preprocessing scripts in a publicly accessible repository. Including reproducible. Although code is included, it's recommended that a Jupyter notebooks for training, evaluation, and ablation would demonstrate commitment to transparency.

Response:

We thank the reviewer for emphasizing the importance of transparency and reproducibility. We confirm that the GitHub link provided in the manuscript currently contains the open-source code required to verify and control our work, including the model architecture, Grad-CAM, and SHAP analysis scripts.

Regarding the pretrained models, weights, and comprehensive notebooks, we have organized these in a private mode, and they will be made fully public immediately upon the manuscript's acceptance at the same repository link provided in the article.

2. Comprehensive Dataset and Method Details. Expand dataset descriptions with detailed information about patient/sample demographics, imaging modalities, annotation standards, and multi-center data harmonization. Include explicit preprocessing and augmentation details to improve reproducibility.

Response:

We appreciate your request for further details regarding the dataset and methodology. We have significantly expanded the 'Dataset and preprocessing' section to explicitly describe the data origins, imaging modalities, and annotation standards. Referencing the foundational work by Tschandl et al. (2018), we have clarified the multi-center nature of the dataset (sourced from Vienna and Queensland) and the harmonization steps taken to standardize images from diverse acquisition systems (e.g., digital dermatoscopy vs. digitized slides). Furthermore, we have ensured that all preprocessing steps (Lines: 331-341) and augmentation hyperparameters (Lines: 357-365) are described in the narrative text to facilitate exact reproducibility. Besides, to ensure exact reproducibility, we have summarized the specific data augmentation hyperparameters in a new table (Table 3).

Changes made to the Manuscript:

• Lines 318–324: We have updated the dataset description to specify the two clinical sources (Vienna and Queensland) and the three imaging modalities (MoleMax HD, DermLite, Diapositives).

• Lines 325–329: We have clarified the ground truth establishment (Histopathology vs. Expert Consensus).

• Lines 365-366, 376: We have added Table 3 to explicitly list the geometric and photometric transformation parameters used during training.

3. Robust Experimental Design and Statistical Rigor. Implement stratified k-fold cross-validation or repeated random splits with detailed fold-wise metrics. Present statistical significance testing (e.g., paired t-test, Wilcoxon) for key results and detailed per-class performance breakdowns.

Response:

We thank the reviewer for this important suggestion regarding methodological rigor. Our study has a specific and novel goal to perform a controlled, comparative analysis of six different fusion architectures that are explicitly designed and evaluated with explainable AI (XAI) integration in mind. By integrating and contrasting two major XAI techniques (Grad-CAM and SHAP), we aim to fill a critical gap in the literature, which often treats architectural performance and post-hoc explainability as separate concerns.

To ensure a fair and interpretable comparison with existing literature and to maintain a consistent basis for our XAI analysis, we deliberately adopted the standard one-hold-out data split of the dataset. This is the same setup used by the majority of studies we have benchmarked against in Table 5 of our manuscript.

Introducing a new k-fold validation protocol at this stage would create an unfair comparison with prior work, as performance metrics are highly sensitive to the specific data partitioning strategy. More critically, it would decouple our core findings from the established benchmark literature, undermining our ability to contextualize the contribution of our XAI-integrated architectural comparisons.

4. Expanded Explainability Approaches. Integrate additional explainability frameworks (e.g., Grad-CAM, LIME, SHAP) with relevant qualitative and quantitative analyses. Include case studies explaining both successes and errors to support clinical interpretability.

Response:

We thank the reviewer for suggesting that we expand our explainability analysis further. We agree that a deeper investigation into model interpretability would significantly strengthen the clinical relevance of our work. In our experiments, we have employed two major complementary XAI frameworks, Grad-CAM (which highlights salient image regions based on gradient flow) and SHAP (which quantifies feature importance based on game theory). We selected these two methods specifically because they follow different fundamental principles. Grad-CAM provides spatial visual explanations, while SHAP offers a quantitative, feature-attribution perspective. This dual-method approach provides a more robust and multifaceted view of model reasoning than using a single technique.

We have added a dedicated subsection titled "Diagnostic Error Analysis and Interpretability Consistency" and a new figure (Fig 13). In this subsection we conducted a granular analysis of misclassified cases from our top-performing model. We identified specific failure modes, including False Positives driven by artifact dependency (e.g., air bubbles) and False Negatives caused by low-contrast lesion boundaries. Moreover, we compared the alignment between Grad-CAM and SHAP.

Changes made to the Manuscript:

• Lines 888–911: Added the new subsection "Diagnostic Error Analysis and Interpretability Consistency" analyzing specific failure modes.

• Figure 13: Added a new figure illustrating a False Positive (Artifact) and a False Negative (Low Contrast) with corresponding Grad-CAM and SHAP visualizations.

5. Ablation and Component Impact Analysis. Provide detailed ablations for model components, data preprocessing steps, and optimization algorithms. Report both quantitative metrics and qualitative results indicating the contribution of each component.

Response:

We thank the reviewer for highlighting the importance of ablation studies to isolate the contribution of individual components. We would like to clarify that the core experimental design of our study is, in essence, a comprehensive and systematic architectural ablation analysis. Our primary objective is to perform a controlled, comparative evaluation of six distinct fusion architectures (our main novel component) against each other and against established CNN and Transformer baselines. Therefore, the detailed comparison presented in Section 4 (Results and Discussion), which reports quantitative metrics (F1-score, Precision, Recall) and qualitative XAI insights (Grad-CAM/SHAP visualizations) for each model, directly serves as an ablation study of the "fusion technique" variable.

6. Broader Benchmarking. Benchmark against state-of-the-art deep learning methods relevant to the problem, including attention-based, transformer, and hybrid architectures. Include efficiency comparisons relevant for clinical use.

Response:

Thanks. Our current experimental design was meticulously crafted to answer the primary research question "How do different fusion strategies between CNNs and ViTs affect performance and, critically, explainability in skin lesion classification?" The side-by-side comparison of six novel fusion architectures on a standardized, challenging benchmark (HAM10000) provides a clear and controlled answer. In Table 5, we have benchmarked our results against relevant recent studies. Adding extensive new comparisons against a broad array of external SOTA models, while informative, would shift the paper's focus from an in-depth architectural investigation to a general benchmarking study. This could dilute the paper's novel message. Furthermore, such comparisons are often hindered by differences in codebase, training protocols, and data splits, making direct, fair comparisons exceptionally complex and potentially misleading without a full re-implementation, which is beyond the scope of this focused contribution.

7. Handling Imbalanced Data and Rare Cases. Discuss strategies employed for imbalanced data or rare conditions, reporting per-class metrics, and experiments with weighted losses or data augmentation.

Response:

We appreciate the reviewer highlighting the issue of class imbalance. In this work, we addressed the uneven distribution of skin lesion classes using a comprehensive data augmentation strategy that targets minority classes. As detailed in the Data augmentation section, we applied specific geometric and photometric transformations to the minority classes to artificially increase their representation in the training set. This approach allows the model to learn from a more diverse set of variations for rare lesion types. We have reported detailed per-class performance metrics, including Precision, Recall, and F1-score, in Table 4. These results demonstrate that our strategy was effective, as evidenced by the high performance on rare classes. For example, FM6 achieved an F1-score of 0.94 for the vascular lesion (VASC) class despite it being a minority category.

8. Clinical Integration and Translation. Elaborate on considerations for real-world deployment in clinical settings, regulatory challenges, device heterogeneity, and ethical implications. Propose plans for external, multicenter, or prospective validation.

Response:

We appreciate this critical feedback. We recognize that high diagnostic accuracy on a retrospective dataset is only the first step towards clinical utility. We have substantially expanded the Integration into clinical workflow section to address the complex translational barriers required for real-world deployment. Specifically, we added a detailed analysis covering three key areas. First, we discussed the challenge of device heterogeneity and the need for domain adaptation to handle images from different hardware. Second, we addressed regulatory considerations for Software as a Medical Device (SaMD), emphasizing the need for explainability to meet transparency standards. Third, we incorporated ethical implications, particularly focusing on the risk of automation bias and the necessity of keeping the human expert in the loop. Furthermore, we refined the Conclusion and future works section to explicitly prioritize external, multi-center, and prospective validation studies to prove robustness beyond the HAM10000 benchmark.

Changes made to the Manuscript:

• Lines 939–953: Added a comprehensive discussion on regulatory, technical, and ethical challenges in the "Integration into clinical workflow" section.

• Lines 994–999: Updated the "Conclusion and future works" section to explicitly propose prospective multi-center validation.

9 Enhanced Literature Integration. Expand literature review to include recent advances and foundational works in biomedical AI, especially explicability, federated learning, and clinical translation, citing:

https://www.sciencedirect.com/science/article/pii/S1476927125000283

https://link.springer.com/article/10.1007/s11831-025-10255-2

https://www.sciencedirect.com/science/article/pii/S1476927125001604?via%3Dihub

https://www.nature.com/articles/s41598-025-12602-6

https://pubmed.ncbi.nlm.nih.gov/40751377/

https://www.nature.com/articles/s41598-025-11574-x

https://link.springer.com/article/10.1007/s11831-025-10315-7

https://www.nature.com/articles/s41598-025-14333-0

https://link.springer.com/article/10.1007/s11831-025-10376-8?utm_source=rct_congratemailt&utm_medium=email&utm_campaign=nonoa_20251002&utm_content=10.1007%2Fs11831-025-10376-8

https://link.springer.com/article/10.1007/s11831-025-10379-5

https://link.springer.com/article/10.1007/s11831-025-10414-5

https://link.springer.com/article/10.1007/s11831-025-10411-8

https://www.sciencedirect.com/science/article/pii/S1746809425013631?r

https://journals.tubitak.gov.tr/biology/vol49/iss5/9/

Response:

We thank the reviewer for providing this comprehensive list of recent and foundational works. We applied a systematic selection criterion to the suggested literature, prioritizing studies that specifically address hybrid feature fusion architectures and Explainable AI in medical imaging. Consequently, we have integrated the publications that align with these methodological scopes into our manuscript. These citations strengthen our discussion on the current state of biomedical AI and the growing necessity for transparent diagnostic tools.

Changes made to the Manuscript:

• Lines 53–61: We have cited the T-FSPANNet, DY-FSPAN, PABT-Net models in the Introduction section. These references illustrate the broader architectural shift in medical imaging towards pyramidal attention and hybrid feature fusion, thereby substantiating the design choices of our proposed framework.

• Lines 221–230: We have added a detailed discussion of the BCB-CSPA network in the Related Works section. We recognize this work as a key benchmark for attention-driven skin lesion classification on the HAM10000 dataset and have discussed how our ViT-Fusion approach complements this direction.

---

## [Decision Letter · Decision Letter 2]

29 Dec 2025

Explainable Depth-Wise and Channel-Wise Fusion Models for Multi-Class Skin Lesion Classification

PONE-D-25-11481R2

Dear Dr. Saleh,

We’re pleased to inform you that your manuscript has been judged scientifically suitable for publication and will be formally accepted for publication once it meets all outstanding technical requirements.

Kind regards,

Lin Xu

Academic Editor

PLOS One

Additional Editor Comments (optional):

Reviewers' comments:

Reviewer's Responses to Questions

**Comments to the Author**

1. If the authors have adequately addressed your comments raised in a previous round of review and you feel that this manuscript is now acceptable for publication, you may indicate that here to bypass the “Comments to the Author” section, enter your conflict of interest statement in the “Confidential to Editor” section, and submit your "Accept" recommendation.

Reviewer #1: All comments have been addressed

Reviewer #3: All comments have been addressed

2. Is the manuscript technically sound, and do the data support the conclusions?

Reviewer #1: Yes

Reviewer #3: Yes

3. Has the statistical analysis been performed appropriately and rigorously?

Reviewer #1: Yes

Reviewer #3: Yes

4. Have the authors made all data underlying the findings in their manuscript fully available?

Reviewer #1: Yes

Reviewer #3: Yes

5. Is the manuscript presented in an intelligible fashion and written in standard English?

Reviewer #1: Yes

Reviewer #3: Yes

6. Review Comments to the Author

Reviewer #1: Dear authors and editors, I have no further concern regarding the revised manuscript. I agree with the adjustments and improvements to the manuscript.

Reviewer #3: (No Response)

7. PLOS authors have the option to publish the peer review history of their article (what does this mean?). If published, this will include your full peer review and any attached files.

Reviewer #1: No

Reviewer #3: No

---

## [Editor Report · Acceptance letter]

PONE-D-25-11481R2

PLOS One

Dear Dr. Saleh,

I'm pleased to inform you that your manuscript has been deemed suitable for publication in PLOS One. Congratulations! Your manuscript is now being handed over to our production team.

Kind regards,

on behalf of

Dr. Lin Xu

Academic Editor

PLOS One